# COHESION: COHERENCE-BASED DIFFUSION FOR LONG-RANGE DYNAMICS FORECASTING

## ABSTRACT

We recast existing works on probabilistic dynamics forecasting through a unified framework connecting turbulence and diffusion principles: Cohesion. Specifically, we relate the coherent part of nonlinear dynamics as a conditioning prior in a denoising process, which can be efficiently estimated using reduced-order models. This fast generation of long prior sequences allows us to reframe forecasting as trajectory planning, a common task in RL. This reformulation is beneficial because we can perform a single conditional denoising pass for an entire sequence, rather than autoregressively over long lead time, gaining orders-of-magnitude speedups with little performance loss. Nonetheless, Cohesion supports flexibility through temporal composition that allows iterations to be performed over smaller subsequences, with autoregressive being a special case. To ensure temporal consistency within and between subsequences, we incorporate a model-free, small receptive window via temporal convolution that leverages large NFEs during denoising. Finally, we perform our guidance in a classifier-free manner to handle a broad range of conditioning scenarios for zero-shot forecasts. Our experiments demonstrate that Cohesion outperforms state-of-the-art probabilistic emulators for chaotic systems over long lead time, including in Kolmogorov Flow and Shallow Water Equation. Its low spectral divergence highlights Cohesion's ability to resolve multi-scale physical structures, even in partially-observed cases, and are thus essential for long-range, high-fidelity, physically-realistic emulation.

## 1 INTRODUCTION

Solving Partial Differential Equations (PDEs) with probabilistic emulators has gained significant momentum relative to their deterministic counterpart Gao et al. (2024a); Rühling Cachay et al. (2024); Gilpin (2024) due to their ability to generate ensemble forecasts that facilitate uncertainty quantification useful for decision making processes Bhatnagar et al. (2019); Brandstetter et al. (2022a;b); Guo et al. (2016); Li et al. (2020); Lu et al. (2021). In particular, diffusion, a powerful class of probabilistic model, has been widely used as emulators in an autoregressive manner to produce sequential forecasts over a target lead time, $\Delta t$ Li et al. (2024); Price et al. (2023); Lippe et al. (2024). However, this probabilistic forecasting approach poses several challenges. First, the conditional denoising process to estimate $p(\mathbf{u} \mid \mathbf{c})$, where $\mathbf{u} \in \mathbb{R}^{n_\mathbf{u}}$ and $\mathbf{c} \in \mathbb{R}^{n_\mathbf{u}}$ are the state and conditioning vector respectively, is generally tied to the condition-generating process. This is primarily due to the use of $(\mathbf{u}, \mathbf{c})$ data pair during training, such that whenever the likelihood $p(\mathbf{c} \mid \mathbf{u})$ changes, one would require further fine-tuning and re-training Stock et al. (2024); Zhao et al. (2024); Gong et al. (2024); Gao et al. (2024a); Chen et al. (2023); Gao et al. (2024b); Hua et al. (2024); Li et al. (2024). Second, a diffusion-based autoregressive approach is extremely costly as one needs to perform multiple denoising passes such that the number of function evaluation (NFEs) grows proportionately with the number of discretization of $\Delta t$ Price et al. (2023); Lippe et al. (2024). This is a problem for many long-range forecasting applications, in weather and climate domains for example, where previous gains in inference speed achieved by deterministic data-driven emulators are quickly offset.

As such, we introduce Cohesion (Figure 1), a diffusion-based forecasting framework that incorporates turbulence and reinforcement learning (RL) principles to achieve accurate and stable long rollouts with orders-of-magnitude inference speedups. First, by leveraging the idea of low-dimensional, coherent flow in turbulence as a conditioning factor within diffusion, we can efficiently generate long

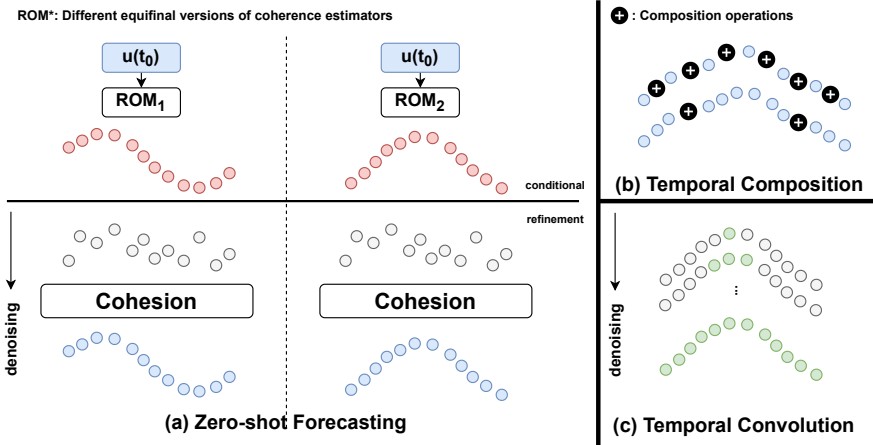

Figure 1: Overview of our Cohesion framework, which reframes forecasting as a trajectory planning task, enabled by a lightweight reduced-order model (ROM) capable of generating conditioning priors efficiently. Key features include: (a) classifier-free guidance for handling broad range of conditioning for zero-shot forecasting; (b) temporal composition through iterative denoising passes which stitch subsequences together; and the use of (c) model-free, small receptive window to ensure local agreement and multi-scale global consistency by exploiting large NFEs during denoising.

sequences of prior $\mathbf{c}$ using a reduced-order model (ROM), such as deep Koopman operator Lusch et al. (2018); Wang et al. (2022). ROMs are especially useful in representing dynamics evolving on low-dimensional attractors dominated by persistent coherent structures Stachenfeld et al. (2021); Solera-Rico et al. (2024), and they demonstrate greater stability over long rollouts compared to high-dimensional models Nathaniel et al. (2024). As a result, we are able to reframe forecasting as trajectory planning – a common task in RL Janner et al. (2022) – which allows us to perform conditional denoising for the entire sequence in a single pass. Nonetheless, Cohesion supports flexibility through temporal composition that allows denoising passes to be performed iteratively over smaller subsequences, with autoregressive being a special case when the subsequence length $R$ is less than the discretization magnitude over $\Delta t$: i.e., $R < T := N_{\Delta t}$. Furthermore, we implement a small receptive window to ensure local agreement at each denoising step, and multi-scale global consistency over the composition of many NFEs without any specialized temporal models Gao et al. (2024a); Rühling Cachay et al. (2024). Finally, we perform our guidance in a classifier-free manner to handle a broad range of conditioning scenarios for zero-shot forecasts.

In order to evaluate Cohesion, we study challenging chaotic spatiotempral dynamics including the Kolmogorov Flow and Shallow Water Equation. For instance, we show that Cohesion is more stable and accurate over state-of-the-art models, including the probabilistic formulation of Spherical Fourier Neural Operator (SFNO) Bonev et al. (2023). Cohesion also has minimal spectral divergences, highlighting its ability to resolve multi-scale structures even in partially-observed cases.

## 2 UNIFIED TURBULENCE-DIFFUSION FRAMEWORK

We begin by recasting existing works on diffusion-based dynamics forecasting through the lens of turbulence theory. Specifically, we consider time-dependent, discrete dynamics across one temporal dimension $t \in [0, T] \subset \mathbb{N}$ and multiple spatial coordinates $\mathbf{x} = [x_1, x_2, \ldots, x_m] \in \mathcal{X}$ (Equation 1).

$$\mathbf{u}(\mathbf{x}, t+1) = \mathcal{F}[\mathbf{u}(\mathbf{x}, t)] \tag{1}$$

where $\mathcal{F} : \mathbb{R}^{n_{\mathbf{u}}} \to \mathbb{R}^{n_{\mathbf{u}}}$ is a differentiable flow map. In general, turbulent dynamics are characterized by a state vector $\mathbf{u}(\mathbf{x}, t)$, and can be represented by a combination of coherent flow and fluctuating component through a mapping operator $\mathcal{H} : \mathbb{R}^{n_{\mathbf{u}}} \to \mathbb{R}^{n_{\mathbf{u}}}$; the linear composition represents the popular Reynolds technique:

$$\mathbf{u}(\mathbf{x}, t) = \mathcal{H}[\ \underbrace{\bar{\mathbf{u}}(\mathbf{x}, t)}_{\text{coherent flow}}\ ,\ \underbrace{\mathbf{u}'(\mathbf{x}, t)}_{\text{fluctuating flow}}\ ] = \underbrace{\bar{\mathbf{u}}(\mathbf{x}, t) + \mathbf{u}'(\mathbf{x}, t)}_{\text{Reynolds decomposition}} \tag{2}$$

Several previous studies have implicitly leveraged the decomposition principle in diffusion-based dynamics forecasting. Here, we make the connection explicit, starting with the coherent flow. Many works, for instance, approximate a variation of $\bar{\mathbf{u}}(\mathbf{x}, t)$ using a deterministic mapping and employ this as a conditioning factor to directly estimate the posteriors over (1) the full solution $p_\phi(\mathbf{u}_K \mid \bar{\mathbf{u}}(\mathbf{x}, t))$, or indirectly through (2) the residual $p_\phi(\mathbf{u}'_K \mid \bar{\mathbf{u}}(\mathbf{x}, t))$. Throughout, the subscripts $\{0, k, K\} \in \mathcal{K}$ refer to the perturbed state vector at the initial, intermediate, and final denoising step, respectively. We explain each of these strategies in turn.

**Coherent flow as conditioning prior**. In order to obtain a conditioning prior within a diffusion framework for forecasting purposes, one often estimates an initial guess for the current timestep using a parameterized model $\mathcal{D} : \mathbb{R}^{n_{\mathbf{u}}} \to \mathbb{R}^{n_{\mathbf{u}}}$, such as $\bar{\mathbf{u}}(\mathbf{x}, t) = \mathcal{D}[\mathbf{u}(\mathbf{x}, t-1)]$ Stock et al. (2024); Zhao et al. (2024); Price et al. (2023); Gong et al. (2024); Gao et al. (2024a); Chen et al. (2023). Others, meanwhile, utilize either a filtered approximation or known system statistics as $\bar{\mathbf{u}}(\mathbf{x}, t)$ Qu et al. (2024); Gao et al. (2024b); Hua et al. (2024); Li et al. (2024). In this work, we define deterministic prior to follow closely with the principle of coherent flow in turbulence theory (more in Section 3.2).

**Full posterior estimation**. Estimating the full posterior solution involves constructing the operator $\mathcal{H}$ based on prior approximation and posterior estimation through an iterative denoising process. This is followed by marginalization over intermediate states, as shown in Equation 3.

$$p_\phi(\mathbf{u}_{0:K} \mid \bar{\mathbf{u}}(\mathbf{x}, t)) := p(\mathbf{u}_0) \prod_{k=1}^{K} p_\phi(\mathbf{u}_k \mid \mathbf{u}_{k-1}, \bar{\mathbf{u}}(\mathbf{x}, t))$$

$$\mathbf{u}(\mathbf{x}, t) \sim p_\phi(\mathbf{u}_K \mid \bar{\mathbf{u}}(\mathbf{x}, t)) = \int p_\phi(\mathbf{u}_{0:K} \mid \bar{\mathbf{u}}(\mathbf{x}, t)) d\mathbf{u}_{0:K-1} \tag{3}$$

Thereafter, probability evaluation is performed, for example by taking an expectation over the conditional posterior at diffusion step $K$ Stock et al. (2024); Zhao et al. (2024); Price et al. (2023); Gong et al. (2024); Gao et al. (2024a); Chen et al. (2023); Qu et al. (2024); Gao et al. (2024b); Hua et al. (2024); Li et al. (2024).

**Residual posterior estimation**. Several works seek to instead estimate the correction term $\mathbf{u}'(\mathbf{x}, t)$, rather than the full solution $\mathbf{u}(\mathbf{x}, t)$ Lippe et al. (2024); Srivastava et al. (2023); Yu et al. (2023); Mardani et al. (2024). Here, $\mathcal{H}$ is first composed of prior approximation. The posterior estimation step in Equation 3 is then followed, but replacing $\mathbf{u}(\mathbf{x}, t) \leftarrow \mathbf{u}'(\mathbf{x}, t)$. After marginalization of intermediate states and posterior evaluation, the residual (or stochastic refinement) is added to the prior, akin to Reynolds linear decomposition (Equation 2 RHS), and shown in Equation 4.

$$\mathbf{u}(\mathbf{x}, t) \approx \underbrace{\bar{\mathbf{u}}(\mathbf{x}, t)}_{\text{deterministic prior}} + \underbrace{\mathbf{u}'(\mathbf{x}, t)}_{\text{stochastic refinement}}\ ;\ \mathbf{u}'(\mathbf{x}, t) \sim p_\phi(\mathbf{u}'_K \mid \bar{\mathbf{u}}(\mathbf{x}, t)) \tag{4}$$

For both approaches, in addition to the different ways for prior approximation, variations exist in terms of how sampling is performed and how post-processing is implemented. After demonstrating the conceptual connection between diffusion and turbulence in the form of coherent-prior and stochastic-refinement pairings, we discuss each component of Cohesion next.

## 3 COHESION: COHERENCE-BASED DIFFUSION

### 3.1 CLASSIFIER-FREE DIFFUSION FOR ZERO-SHOT FORECASTING

**Forward diffusion**. At each time step in the forward diffusion process, a sample $\mathbf{u} \sim p(\mathbf{u})$ is progressively perturbed through a continuous diffusion timestepping. This process is described by a

linear stochastic differential equation (SDE), as shown in Equation 5 Song et al. (2020).

$$d\mathbf{u}_k = \underbrace{f(k)\mathbf{u}_k\,dk}_{\text{drift term}} + \underbrace{g(k)\,dw(k)}_{\text{diffusion term}} \tag{5}$$

where $f(k)$ and $g(k) \in \mathbb{R}$ are the drift and diffusion coefficients. Here, $w(k) \in \mathbb{R}^{n_\mathbf{u}}$ represents a Wiener process (standard Brownian motion), and $\mathbf{u}_k \in \mathbb{R}^{n_\mathbf{u}}$ denotes the perturbed sample at diffusion step $k \in [0, K = 1] \subset \mathbb{R}$. We use `cosine` noise scheduler in variance-preserving (VP) SDE Nichol & Dhariwal (2021); Chen (2023).

**Reverse denoising**. The reverse denoising process is represented by a reverse SDE as defined in Equation 6 Song et al. (2020), where the score function is approximated with a learnable score network, $s_\theta(\mathbf{u}_k, k)$. The objective function would be to minimize a continuous weighted combination of Fisher divergences between $s_\theta(\mathbf{u}_k, k)$ and $\nabla_{\mathbf{u}_k} \log p(\mathbf{u}_k)$ through score matching Vincent (2011); Song et al. (2020).

$$d\mathbf{u}_k = [\underbrace{f(k)\mathbf{u}_k}_{\text{drift term}} - g(k)^2 \underbrace{\nabla_{\mathbf{u}_k} \log p(\mathbf{u}_k)}_{\text{score function}}]dk + \underbrace{g(k)dw(k)}_{\text{diffusion term}} \tag{6}$$

However, the perturbed state distribution $p(\mathbf{u}_k)$ is data-dependent and unscalable. As such, we reformulate the objective function by replacing $\nabla_{\mathbf{u}_k} \log p(\mathbf{u}_k)$ with $\nabla_{\mathbf{u}_k} \log p(\mathbf{u}_k \mid \mathbf{u})$ where the analytical form of the perturbation kernel is accessible Vincent (2011). In order to improve the stability of the objective, especially closer to the start of the denoising step ($k \to 0$), we apply a reparameterization trick which replaces $s_\theta(\mathbf{u}_k, k) = -\epsilon_\theta(\mathbf{u}_k, k)/\sigma(k)$, where $\Sigma = \sigma^2$ as in Equation 7 Zhang & Chen (2022).

$$\min_\theta \mathbb{E}_{p(\mathbf{u}), p(k), p(\epsilon) \sim \mathcal{N}(0, \mathbf{I})} \left[ \|\epsilon_\theta(\mu(k)\mathbf{u} + \sigma(k)\epsilon, k) - \epsilon)\|_2^2 \right] \tag{7}$$

Following standard convention, we denote $\epsilon_\theta(\mathbf{u}_k, k)$ with $s_\theta(\mathbf{u}_k, k)$ for cleaner notation.

**Zero-shot conditional sampling**. The case we have discussed so far is the unconditional sampling process as we try to sample $\mathbf{u} \sim p(\mathbf{u}_K)$. In order to condition the generative process with $\mathbf{c} := \bar{\mathbf{u}}(\mathbf{x}, t)$, we seek to sample from $\mathbf{u} \sim p(\mathbf{u}_K \mid \mathbf{c})$. This can be done by modifying the score as in Equation 6 with $\nabla_{\mathbf{u}_k} \log p(\mathbf{u}_k \mid \mathbf{c})$ and plugging it back to the reverse SDE process.

As noted earlier, however, one would need fine-tuning or re-training whenever the observation process $p(\mathbf{c} \mid \mathbf{u})$ changes. Nonetheless, several works have attempted to approximate the conditional score with just a single pre-trained network, bypassing the need for expensive re-training Song et al. (2020); Chung et al. (2022). First, using Bayes rule, we expand the conditional score as:

$$\nabla_{\mathbf{u}_k} \log p(\mathbf{u}_k \mid \mathbf{c}) = \underbrace{\nabla_{\mathbf{u}_k} \log p(\mathbf{u}_k)}_{\text{unconditional score}} + \underbrace{\nabla_{\mathbf{u}_k} \log p(\mathbf{c} \mid \mathbf{u}_k)}_{\text{log-likelihood function}} \tag{8}$$

Since the first term on the right-hand side is already approximated by the unconditional score network, the remaining task is to identify the second log-likelihood function. Assuming a Gaussian observation process, the approximation goes as in Equation 9 Chung et al. (2022).

$$p(\mathbf{c} \mid \mathbf{u}_k) = \int p(\mathbf{c} \mid \mathbf{u})p(\mathbf{u} \mid \mathbf{u}_k)d\mathbf{u} \approx \mathcal{N}(\mathbf{c} \mid \hat{\mathbf{u}}(\mathbf{u}_k), \sigma_\mathbf{c}^2) \tag{9}$$

The mean $\hat{\mathbf{u}}(\mathbf{u}_k)$ can be approximated by the Tweedie's formula Efron (2011) as in Equation 10.

$$\hat{\mathbf{u}}(\mathbf{u}_k) \approx \frac{\mathbf{u}_k + \sigma^2(k)s_\theta(\mathbf{u}_k, k)}{\mu(k)} \tag{10}$$

Following works from Rozet & Louppe (2023); Qu et al. (2024), we improve the numerical stability by injecting information about the noise-signal ratio in the variance term, i.e., $\sigma_\mathbf{c}^2 + \gamma[\sigma^2(k)/\mu^2(k)]\mathbf{I}$,

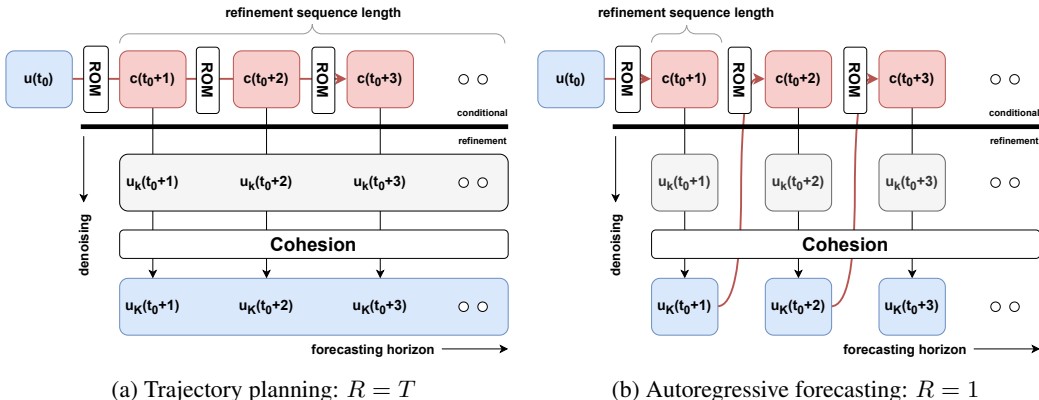

(a) Trajectory planning: $R = T$        (b) Autoregressive forecasting: $R = 1$

Figure 2: By temporal composition, we allow for flexible refinement subsequence size $R$. (a) Trajectory planning approach ($R = T$ case) where only one pass of conditional denoising is performed. (b) Autoregressive forecasting approach ($R = 1$ case) requires multiple backward passes.

where $\gamma$, $\mathbf{I}$ are scalar constant and the identity matrix respectively. We now have a classifier-free posterior diffusion sampling where $\nabla \mathbf{u}_k \log p(\mathbf{u}_k \mid \mathbf{c})$ can be approximated using a single unconditional score network $s_\theta(\mathbf{u}_k, k)$, allowing for zero-shot forecasts given different conditioning scenarios (see Algorithm 1).

**Predictor-corrector**. We implement a predictor-corrector procedure to enhance the quality of our conditional generative process Song et al. (2020). The reverse SDE prediction process is solved using the exponential integrator (EI) discretization scheme as in Equation 11 Zhang & Chen (2022). The correction phase employs several steps of Langevin Monte Carlo (LMC) to adjust for discretization errors, utilizing a sufficiently small Langevin amplitude $\tau \in \mathbb{R}_+$ as in Equation 12 Song et al. (2020) (see Algorithm 2).

$$\mathbf{u}_{k+\Delta k} \leftarrow \frac{\mu(k + \Delta k)}{\mu(k)} \mathbf{u}_k + \left( \frac{\mu(k + \Delta k)}{\mu(k)} + \frac{\sigma(k + \Delta k)}{\sigma(k)} \right) \Sigma(k) s_\theta(\mathbf{u}_k, k \mid \mathbf{c}) \tag{11}$$

$$\mathbf{u}_k \leftarrow \mathbf{u}_k + \tau s_\theta(\mathbf{u}_k, k) + \sqrt{2\tau}\epsilon. \tag{12}$$

## 3.2 LEARNING COHERENT STRUCTURES

Koopman theory Koopman & Neumann (1932) demonstrates that nonlinear dynamics can be modeled by an infinite-dimensional linear Koopman operator acting on the space of all possible measurement functions. Leveraging a deep encoder-decoder model, $\{\mathcal{G}_E, \mathcal{G}_D\} \in \mathcal{G}$ Lusch et al. (2018), the Koopman operator $\mathcal{O} : \mathcal{G}_E(\mathcal{X}) \mapsto \mathcal{G}_E(\mathcal{X})$ acts on a lower ($n_\mathbf{d}$)-dimensional latent manifold that advances the state vector in time (see Equation 13).

$$\mathcal{O}[\mathcal{G}_E(\mathbf{u}(\mathbf{x}, t))] := \mathcal{G}_E \circ \mathcal{F}[\mathbf{u}(\mathbf{x}, t)] = \mathcal{G}_E \circ \mathbf{u}(\mathbf{x}, t + 1) \tag{13}$$

A conditioning prior is then generated by the decoder as:

$$\bar{\mathbf{u}}(\mathbf{x}, t + 1) := \mathcal{G}_D \circ \mathcal{O}[\mathcal{G}_E(\mathbf{u}(\mathbf{x}, t))] \tag{14}$$

Composing $\mathcal{O}$ for $m$ times within Equation 14 results in the generation of an autoregressive sequence of conditioning priors that extends over $m$ steps. We perform joint training by minimizing the 1-step lagged reconstruction loss as in Equation 15. We collectively refer to $\{\mathcal{G}_E, \mathcal{O}, \mathcal{G}_D\} \in f_\psi$ as the reduced-order model (ROM), where $f_\psi : \mathbb{R}^{n_\mathbf{u}} \rightarrow \mathbb{R}^{n_\mathbf{u}}$.

$$\min_{\{\mathcal{G}_E, \mathcal{O}, \mathcal{G}_D\} \in f_\psi} \mathbb{E}_{p(\mathbf{u})} \left[ \| \bar{\mathbf{u}}(\mathbf{x}, t + 1) - \mathbf{u}(\mathbf{x}, t + 1) \|_2^2 \right] \tag{15}$$

### 3.3 FORECASTING AS TRAJECTORY PLANNING

When a denoising pass is performed iteratively, as evidenced in many autoregressive tasks Price et al. (2023); Srivastava et al. (2023), the computational costs can become prohibitively expensive. By leveraging our proposed compute-efficient ROM $f_\psi$, we can generate a sequence of conditioning priors $\mathcal{C}(\mathbf{x}) \in \mathbb{R}^{R \times n_\mathbf{u}}$ of length $R$ as:

$$\mathcal{C}(\mathbf{x}) = \{\mathbf{c}(t_0 + 1) := f_\psi^1(\mathbf{u}(\mathbf{x}, t_0)), \cdots, \mathbf{c}(t_0 + R) := f_\psi^R(\mathbf{u}(\mathbf{x}, t_0))\}_{1:R} \quad (16)$$

where $\mathbf{u}(\mathbf{x}, t_0)$ is the initial condition. We then perform conditional denoising given $\mathcal{C}(\mathbf{x})$ to estimate $\mathcal{U}(\mathbf{x}) \in \mathbb{R}^{R \times n_\mathbf{u}}$ as in Equation 17.

$$\mathcal{U}(\mathbf{x}) = \{\mathbf{u}(\mathbf{x}, t_0 + 1), \cdots, \mathbf{u}(\mathbf{x}, t_0 + R)\}_{1:R} \sim p_\phi(\mathcal{U}_K(\mathbf{x}) \mid \mathcal{C}(\mathbf{x})) \quad (17)$$

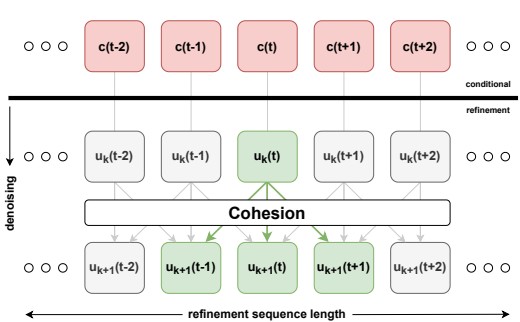

Figure 3: A single denoising step with local receptive window of size $W = 3$. Multiple composition through many NFEs ensures global consistency.

Figure 2a illustrates an example of trajectory planning ($R = T$), with a single denoising pass. We also provide flexibility and allow $R \in [1, T] \subset \mathbb{N}$. In the case where $R < T$, we can simply repeat the denoising passes $r = 1 : \lceil T/R \rceil$ times, where $\lceil . \rceil$ is the ceiling operation, and $\mathcal{C}(\mathbf{x})$ is generated using the previous-step forecast as the initial condition whenever $r > 1$. Figure 2b illustrates this case for $R = 1$, a special case for the classic next-step autoregressive approach. In order to effectively capture multi-scale temporal information, we also incorporate a model-free local receptive window of size $W \in [1, R] \subset \mathbb{N}$. This approach ensures local agreement during each conditional denoising step by training the score model on $W$-length subsequences (see Algorithm 3). By composing many such steps during inference (see Algorithm 4), local agreement translates to global consistency. Figure 3 shows a single temporal convolution during a single denoising step illustrating a window size of $W = 3$ captures local context. For this work, we use $W = 5$ during the training of and sampling using the score network.

## 4 EXPERIMENTS

**Baselines**. We use probabilistic Spherical Fourier Neural Operator (SFNO) Bonev et al. (2023) as a baseline, building on FNO Li et al. (2020), which leverages Fast Fourier and Spherical Harmonic Transforms (SHT) to model Earth's fluid dynamics, including Kolmogorov Flow and the Shallow Water Equation used in this study. We use the off-the-shelf SFNO implementation[1], widely employed in weather Kurth et al. (2023) and climate emulation Watt-Meyer et al. (2023). To ensure a fair comparison, we scale SFNO's parameters to match or exceed those of Cohesion and introduce probabilistic modifications. Unless stated, all models are evaluated on five samples/members.

- **Checkpoints**: Ensembles through multiple model fitting initialized randomly.
- **MC-Dropout**: Ensembles by enabling inference-time dropouts.
- **IC Perturbation**: Ensembles through the perturbation of initial conditions.

**Metrics**. In addition to pixel-based (RMSE, MAE; Equations 19-20) and structure-based (MS-SSIM; Equation 25) metrics, we also use physics-based metrics of spectral divergence evaluated at the final forecasting step, $\Delta t$ (Equation 31). The latter is especially crucial to measure how well multi-scale structures are preserved. A smooth model (i.e., low fidelity) can perform better on metrics like RMSE, but poorly on the spectral domain in e.g., capturing high-frequency signal. The notation (R=1) and (R=T) indicate Cohesion in either autoregressive or trajectory planning mode.

---

[1]https://github.com/NVIDIA/torch-harmonics

### 4.1 KOLMOGOROV FLOW

Incompressible fluid dynamics are governed by the Navier-Stokes equations:

$$\dot{\mathbf{u}} = -(\mathbf{u} \cdot \nabla)\mathbf{u} + \frac{1}{\text{Re}}\nabla^2\mathbf{u} - \frac{1}{\rho}\nabla p + \mathbf{f},$$

$$0 = \nabla \cdot \mathbf{u} \tag{18}$$

where $\mathbf{u}$ is the velocity field, $\text{Re} = 10^3$ is the Reynolds number, $\rho = 1$ is the fluid density, $p$ is the pressure field, and $\mathbf{f}$ is the external forcing. Following Kochkov et al. (2021) and using `jax-cfd`[2] as solvers, we consider a two-dimensional domain $[0, 2\pi]^2$ with periodic boundary conditions and an external forcing $\mathbf{f}$ corresponding to Kolmogorov forcing with linear damping.

**Experimental setup**. The Navier-Stokes Equations 18 are solved on a $256{\times}256$ grid, downsampled to a $64{\times}64$ resolution, with an integration time step of $\Delta = 0.2$ model time units between successive snapshots of the velocity field $\mathbf{u}$. We generated 8196 independent trajectories – each of length 64 and discarding the first half of warm-ups – subsequently dividing them into `80-10-10` train-val-test trajectory-level split. More details in Appendix C.1.

**Model architectures**. The ROM consists of 5 symmetrical convolution layers in the $\mathcal{G}_E - \mathcal{O} - \mathcal{G}_D$ composition with hidden size of $[4, 8, 16, 32, 64]$ and embedding dimension of $n_{\mathbf{d}} = 64$. The score network is parameterized by modern U-Net with $[3, 3, 3]$ residual blocks He et al. (2016), each consisting of $[16, 32, 64]$ hidden channels. The temporal component of the score network is parameterized by a two-layer dense network with 256 hidden channels and 64-dimensional embedding.

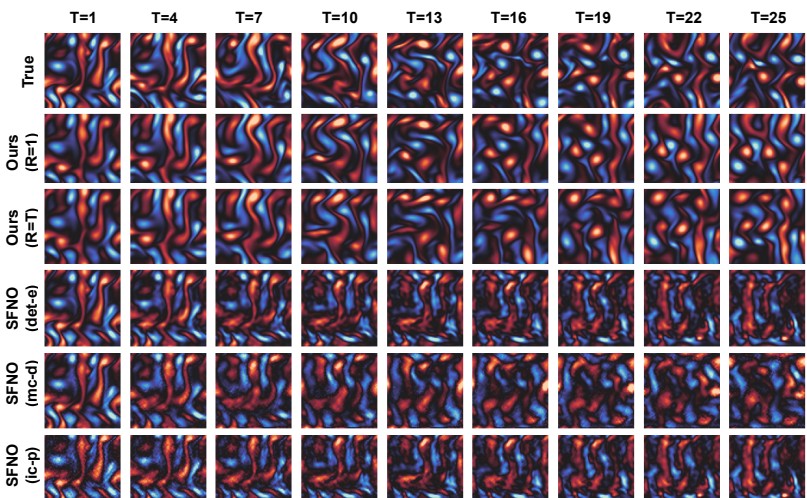

Figure 4: Qualitative result for Kolmogorov Flow where Cohesion is stable and able to capture fine details over long rollouts compared to its probabilistic baselines.

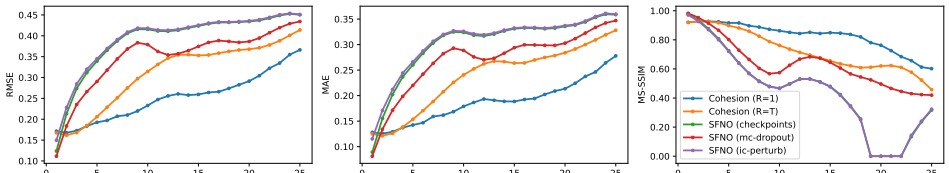

Figure 5: Quantitative result for Kolmogorov Flow where Cohesion has the lowest RMSE ($\downarrow$), MAE ($\downarrow$), and highest MS-SSIM ($\uparrow$) over long rollouts compared to its probabilistic baselines.

---

[2]https://github.com/google/jax-cfd

**Results**. As illustrated in Figure 4, we demonstrate that Cohesion is able to capture fine details over long rollouts. This is also highlighted by Cohesion's ability to outperform probabilistic baselines in the pixel-based, structure-based metrics (Figure 5), as well as physics-based scores (Figure 8a).

## 4.2 SHALLOW WATER EQUATION (SWE)

The SWE system can be described by a set of nonlinear hyperbolic PDEs that governs the dynamics of thin-layer "shallow" fluid where its depth is negligible relative to the characteristic wavelength. Thus, SWE is ideal to model planetary fluid phenomena Bonev et al. (2018).

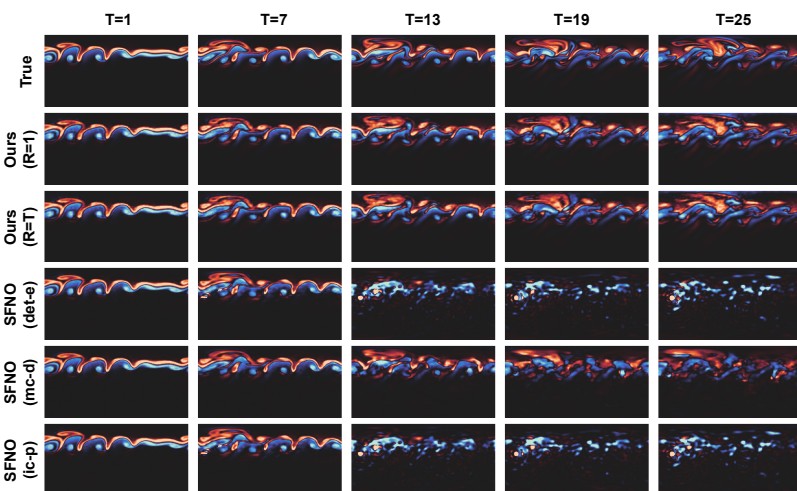

Figure 6: Qualitative result for SWE where Cohesion is stable and able to capture fine details over long rollouts compared to its probabilistic baselines.

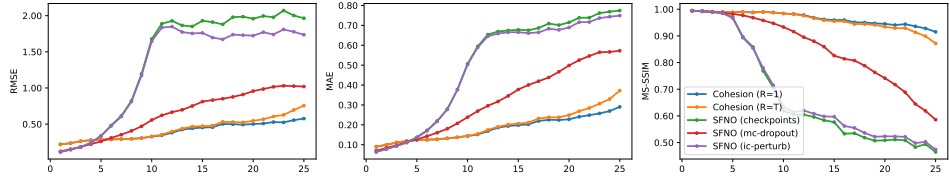

Figure 7: Quantitative result for SWE where Cohesion has the lowest RMSE ($\downarrow$), MAE ($\downarrow$), and highest MS-SSIM ($\uparrow$) over long rollouts compared to its probabilistic baselines.

**Experimental setup**. We generate 2048 trajectories of SWE on a rotating sphere Bonev et al. (2023), with a $80\text{-}10\text{-}10$ train-val-test trajectory-level split. Each trajectory is randomly initialized with an average geopotential height of $\varphi_{avg} = 10^3 \cdot g$ and a standard deviation $\varphi_{amp} \sim \mathcal{N}(120, 20) \cdot g$, on a Galewsky setup to mimic barotropically unstable mid-latitude jet Galewsky et al. (2004). The spatial resolution is $120 \times 240$, keeping the last $\Delta t = N_{\Delta t} = 32$ of vorticity snapshots. More details in Appendix C.2.

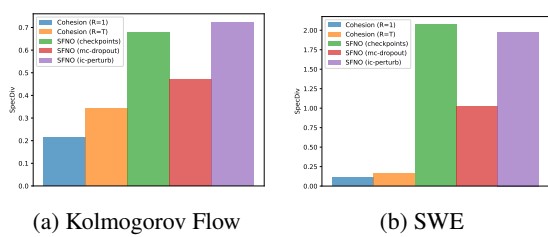

(a) Kolmogorov Flow       (b) SWE

Figure 8: Cohesion has the lowest spectral divergence ($\downarrow$) across probabilistic baselines, indicating its ability to capture multi-scale physical structures.

**Model architectures**. The ROM consists of 5 symmetrical convolution layers in the $\mathcal{G}_E - \mathcal{O} - \mathcal{G}_D$ composition with hidden size of [8, 16, 32, 64, 128] and embedding dimension of $n_{\mathbf{d}} = 128$. The score network is setup identically with that in Kolmogorov Flow.

**Results**. We first demonstrate that Cohesion is able to capture fine details over long rollouts (Figure 6). This is also highlighted by Cohesion's ability to outperform probabilistic baselines in the pixel-based, structure-based metrics (Figure 7), as well as physics-based scores (Figure 8b).

## 4.3 COHESION AS A PHYSICALLY-CONSISTENT PROBABILISTIC EMULATOR

We further demonstrate Cohesion's ability to generate physically-consistent forecasts over long rollouts, even in partially-observed cases. Evaluations are based on Cohesion as trajectory planner.

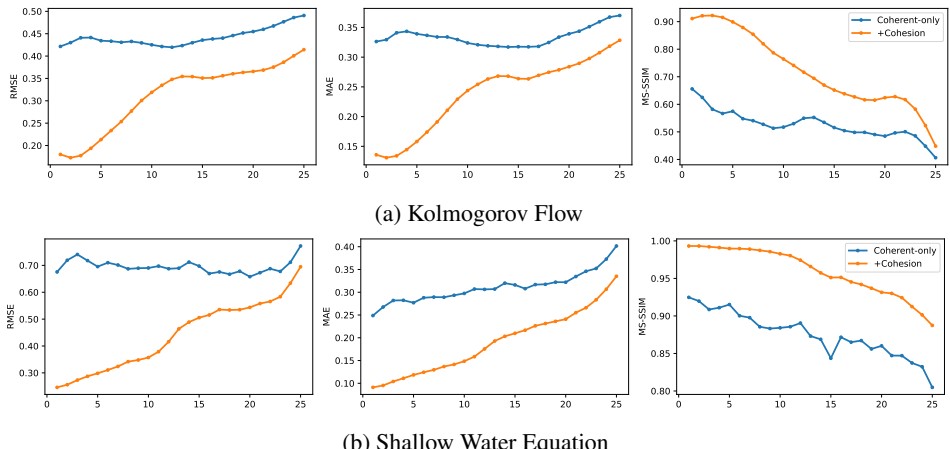

Figure 9: Cohesion as a refiner: Cohesion improves RMSE ($\downarrow$), MAE ($\downarrow$), and MS-SSIM ($\uparrow$) scores over its coherent-only prior forecasts generated sequentially with ROM.

**Cohesion as a refiner**. As shown in Figure 9, Cohesion acts as a refiner of prior forecasts generated by ROM. While ROM provide fast approximations of the system's evolution, they often lack the resolution necessary to capture fine details of the system, particularly in complex chaotic flows. Cohesion enhances these coarse forecasts by applying a diffusion-based refinement process, improving their alignment with high-fidelity simulation.

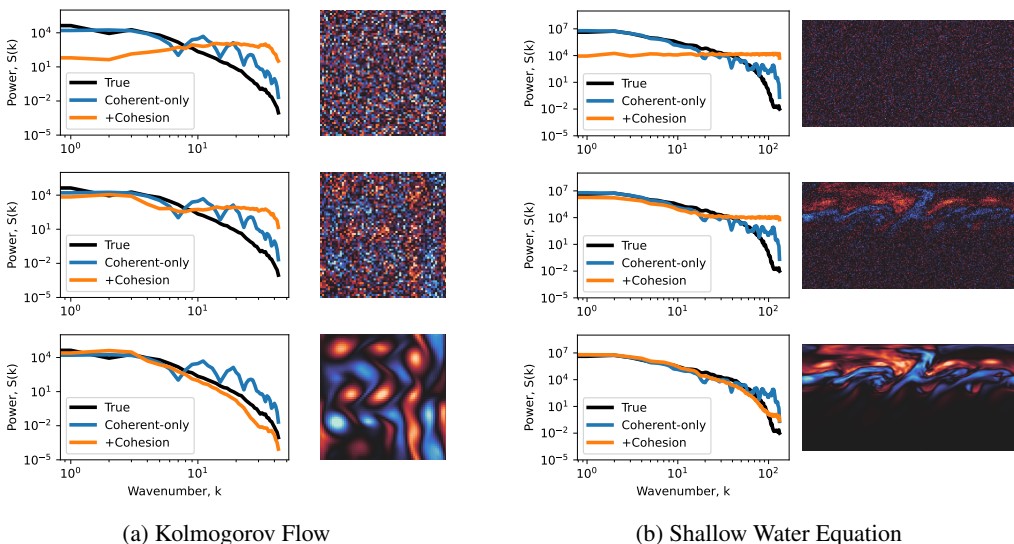

Figure 10: Cohesion as a resolver: Cohesion resolves multi-scale physics ubiquitous in chaotic dynamics even after long rollouts (T = $\Delta t$), by first getting accurate coherent flow i.e., low-frequency signal ($\downarrow k$), before correcting for the fluctuating component i.e., high-frequency signal ($\uparrow k$). Top-middle-bottom rows represent initial-middle-final denoising steps.

**Cohesion as a resolver**. As shown in Figure 10 (evaluated at T = $\Delta t$), Cohesion also resolves multi-scale physics where it first captures low-frequency signals (low wavenumber, $k$), which correspond to the coherent features of the system (e.g., dominant wave pattern). Once the coherent flow is well-represented, Cohesion then resolves high-frequency signals (high wavenumber, $k$), which relate to the faster-evolving turbulent features (e.g., eddies) that arise from coupled nonlinearity.

**Cohesion is physically grounded**. As shown in Figure 11, Cohesion generates high-resolution, realistic physics even in the presence of partially observed conditioning priors. This property is essential for modeling real-world dynamics where priors may be incomplete (e.g., due to sparse representation) or inconsistent (e.g., due to system biases), enabling the framework to handle uncertainty over long unrolls in a physically-grounded manner.

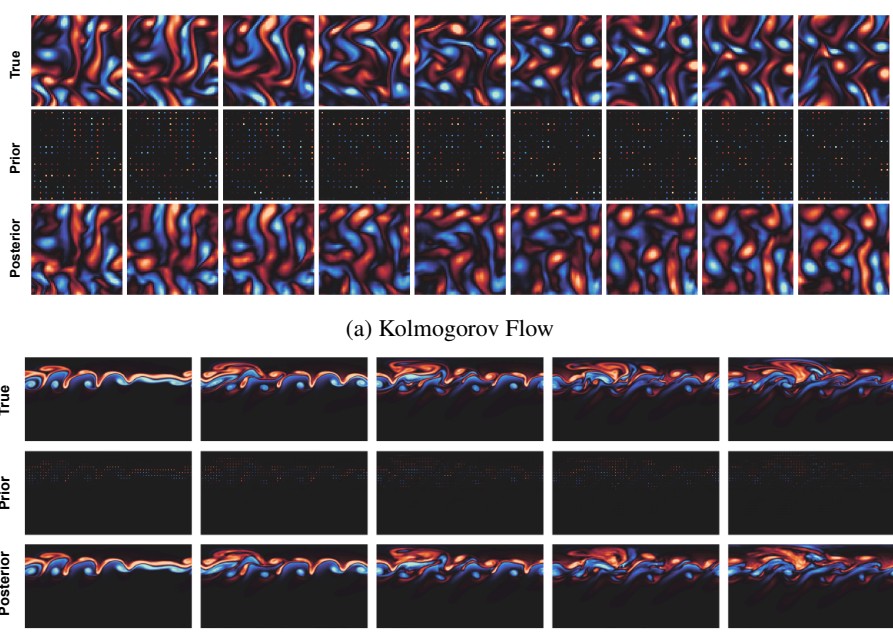

(a) Kolmogorov Flow

(b) Shallow Water Equation

Figure 11: Cohesion produces physically-consistent and realistic forecasts at long unrolls even in the presence of partially-observed conditioning prior. In this experiment, we apply equally-spaced masking to the coherent dynamics generated by ROM, which is then used as a conditioning prior during the denoising process.

## 5 CONCLUSION

We introduce Cohesion, a diffusion-based forecasting framework developed with turbulence and RL principles that is both cheap and achieves stable, accurate, and realistic long simulation rollouts. By reframing autoregressive forecasting as trajectory planning, we gain significant speedups (Figure 12) while maintaining performance. This is enabled by reduced-order modeling, temporal composition, and temporal convolution to ensure multi-scale, local-global consistency. Our extensive examination of Cohesion on Kolmogorov Flow and Shallow Water Equation, in terms of improved performance over state-of-the-art probabilistic emulator across metrics presents an important step toward resolving multi-scale physics in an efficient manner, even in partially-observed cases. This approach can extend predictability and improve the fidelity and realism of data-driven emulators for chaotic systems, like weather and climate, leading to actionable insights.

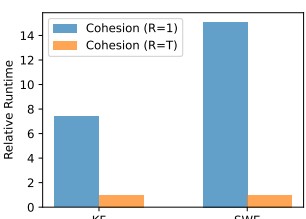

Figure 12: Relative inference runtime in R=1 and R=T settings with identical resource.

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

## A  ALGORITHMS

Some of these algorithms are inspired by, and extended from Rozet & Louppe (2023) and Janner et al. (2022), though these are set to solve different problems.

---

**Algorithm 1** Posterior score estimation, $\nabla_{\mathbf{u}_k} \log p(\mathbf{u}_k \mid \mathbf{c})$

---

1: **function** POSTERIORESTIMATE$(s_\theta, \mathbf{u}_k, k, \mathbf{c})$
2:     $s_{\mathbf{u}} \leftarrow s_\theta(\mathbf{u}_k, k)$
3:     $\hat{\mathbf{u}} \leftarrow \frac{\mathbf{u}_k + \sigma^2(k) s_{\mathbf{u}}}{\mu(k)}$
4:     $s_{\mathbf{c}} \leftarrow \nabla_{\mathbf{u}_k} \log \mathcal{N}(\mathbf{c} \mid \hat{\mathbf{u}}, \sigma_{\mathbf{c}}^2 + \gamma \frac{\sigma^2(k)}{\mu^2(k)} \mathbf{I})$
5:     **return** $s_{\mathbf{u}} + s_{\mathbf{c}}$
6: **end function**

---

**Algorithm 2** Predictor-corrector sampling

---

1: **function** DIFFUSIONSAMPLING$(s_\theta, \tau, N_c)$
2:     $\mathbf{u}_{\mathcal{K}_0 = 0} \sim \mathcal{N}(0, \sigma^2(\mathbf{I}))$
3:     **for** $i = 0$ to $|\mathcal{K}|$ **do**
4:         $s_p \leftarrow$ PosteriorEstimate$(s_\theta, \mathbf{u}_{\mathcal{K}_i}, \mathcal{K}_i, \mathbf{c})$                   ▷ see Algorithm 1
5:         $\mathbf{u}_{\mathcal{K}_{i+1}} \leftarrow \frac{\mu(\mathcal{K}_{i+1})}{\mu(\mathcal{K}_i)} \mathbf{u}_{\mathcal{K}_i} + \left( \frac{\mu(\mathcal{K}_{i+1})}{\mu(\mathcal{K}_i)} - \frac{\sigma(\mathcal{K}_{i+1})}{\sigma(\mathcal{K}_i)} \right) \sigma^2(\mathcal{K}_i) s_p$                   ▷ Predictor
6:         **for** $j = 0$ to $N_c$ **do**
7:             $\epsilon \sim \mathcal{N}(0, \mathbf{I})$
8:             $s_c \leftarrow$ PosteriorEstimate$(s_\theta, \mathbf{u}_{\mathcal{K}_{i+1}}, \mathcal{K}_{i+1}, \mathbf{c})$                   ▷ see Algorithm 1
9:             $\mathbf{u}_{\mathcal{K}_{i+1}} \leftarrow \mathbf{u}_{\mathcal{K}_{i+1}} + \tau s_c + \sqrt{2\tau}\epsilon$                   ▷ Corrector
10:         **end for**
11:     **end for**
12:     **return** $\mathbf{u}_K$
13: **end function**

---

**Algorithm 3** Score network training with window of size, $W$

---

**Require:** $W \mod 2 = 1$                   ▷ Symmetric window about $\mathbf{u}(t_i)$
1: $W \leftarrow 2w + 1$
2: **while** not done **do**
3:     $i \sim \mathcal{U}(\{w + 1, \cdots, T - w\})$
4:     $k \sim \mathcal{U}(\mathcal{K}), \epsilon \sim \mathcal{N}(0, \mathbf{I})$
5:     $\mathbf{u}_k(t_{i-w:i+w}) \leftarrow \mu(k)\mathbf{u}(t_{i-w:i+w}) + \sigma(k)\epsilon$
6:     Loss $\leftarrow \|\epsilon_\theta(\mathbf{u}_k(t_{i-w:i+w}), k) - \epsilon\|_2^2$
7:     $\theta \leftarrow$ GradientUpdate$(\theta, \nabla_\theta$Loss$)$
8: **end while**

---

**Algorithm 4** Temporal convolution with local receptive window within (sub)sequences

---

**Require:** $1 \leq w \leq R$
1: **function** TEMPORALCONVOLUTION$(s_\theta, \mathbf{u}_k, \mathbf{c}, k, w, R)$
2:     $s_{1:w+1} \leftarrow s_\theta(\mathbf{u}_k(t_{1:2w+1}), k \mid \mathbf{c})[: w + 1]$
3:     **for** $i = w + 2$ to $R - w - 1$ **do**
4:         $s_i \leftarrow s_\theta(\mathbf{u}_k(t_{i-w:i+w}), k \mid \mathbf{c})[w + 1]$
5:     **end for**
6:     $s_{R-w:R} \leftarrow s_\theta(\mathbf{u}_k(t_{R-2w:R}), k \mid \mathbf{c})[w + 1 :]$
7:     **return** $s_{1:R}$
8: **end function**

---

## B    METRICS

We divide our metrics into pixel-based, structure-based, and physics-based. The former two deal with information loss in the data space, while the latter in the spectral space.

### B.1    PIXEL-BASED METRICS

As described in the main text, we apply root mean-squared error (RMSE; equation 19) and mean absolute error (MAE; equation 20)

$$\mathcal{M}_{RMSE} = \sqrt{\frac{1}{n_{\mathbf{u}}} \sum (\hat{\mathbf{u}} - \mathbf{u})^2} \tag{19}$$

$$\mathcal{M}_{MAE} = \frac{1}{n_{\mathbf{u}}} \sum |\hat{\mathbf{u}} - \mathbf{u}| \tag{20}$$

### B.2    STRUCTURE-BASED METRICS

Let $\mathbf{Y}$ and $\hat{\mathbf{Y}}$ be two images to be compared, and let $\mu_{\mathbf{Y}}$, $\sigma_{\mathbf{Y}}^2$ and $\sigma_{\mathbf{Y}\hat{\mathbf{Y}}}$ be the mean of $\mathbf{Y}$, the variance of $\mathbf{Y}$, and the covariance of $\mathbf{Y}$ and $\hat{\mathbf{Y}}$, respectively. The luminance, contrast and structure comparison measures are defined as follows:

$$l(\mathbf{Y}, \hat{\mathbf{Y}}) = \frac{2\mu_{\mathbf{Y}}\mu_{\hat{\mathbf{Y}}} + C_1}{\mu_{\mathbf{Y}}^2 + \mu_{\hat{\mathbf{Y}}}^2 + C_1}, \tag{21}$$

$$c(\mathbf{Y}, \hat{\mathbf{Y}}) = \frac{2\sigma_{\mathbf{Y}}\sigma_{\hat{\mathbf{Y}}} + C_2}{\sigma_{\mathbf{Y}}^2 + \sigma_{\hat{\mathbf{Y}}}^2 + C_2}, \tag{22}$$

$$s(\mathbf{Y}, \hat{\mathbf{Y}}) = \frac{\sigma_{\mathbf{Y}\hat{\mathbf{Y}}} + C_3}{\sigma_{\mathbf{Y}}\sigma_{\hat{\mathbf{Y}}} + C_3}, \tag{23}$$

where $C_1$, $C_2$ and $C_3$ are constants given by

$$C_1 = (K_1 L)^2, C_2 = (K_2 L)^2, \text{ and } C_3 = C_2/2. \tag{24}$$

$L = 255$ is the dynamic range of the gray scale images, and $K_1 \ll 1$ and $K_2 \ll 1$ are two small constants. To compute the MS-SSIM metric across multiple scales, the images are successively low-pass filtered and down-sampled by a factor of 2. We index the original image as scale 1, and the desired highest scale as scale $M$. At each scale, the contrast comparison and structure comparison are computed and denoted as $c_j(\mathbf{Y}, \hat{\mathbf{Y}})$ and $s_j(\mathbf{Y}, \hat{\mathbf{Y}})$ respectively. The luminance comparison is only calculated at the last scale $M$, denoted by $l_M(\mathbf{Y}, \hat{\mathbf{Y}})$. Then, the MS-SSIM metric is defined by

$$\mathcal{M}_{MS-SSIM} = [l_M(\mathbf{Y}, \hat{\mathbf{Y}})]^{\alpha_M} \cdot \prod_{j=1}^{M} [c_j(\mathbf{Y}, \hat{\mathbf{Y}})]^{\beta_j} [s_j(\mathbf{Y}, \hat{\mathbf{Y}})]^{\gamma_j} \tag{25}$$

where $\alpha_M$, $\beta_j$ and $\gamma_j$ are parameters. We use the same set of parameters as in Wang et al. (2003): $K_1 = 0.01$, $K_2 = 0.03$, $M = 5$, $\alpha_5 = \beta_5 = \gamma_5 = 0.1333$, $\beta_4 = \gamma_4 = 0.2363$, $\beta_3 = \gamma_3 = 0.3001$, $\beta_2 = \gamma_2 = 0.2856$, $\beta_1 = \gamma = 0.0448$. The predicted and ground truth images of physical variables are re-scaled to 0-255 prior to the calculation of their MS-SSIM values.

### B.3    PHYSICS-BASED METRICS

We next describe in detail the definition and implementation of our physics-based metrics, particularly Spectral Divergence (SpecDiv). Consider a 2D image field of size $h \times w$ for a physical parameter at a specific time, variable, and level. Let $f(x, y)$ be the intensity of the pixel at position $(x, y)$. First, we compute the 2D Fourier transform of the image by:

$$F(k_x, k_y) = \sum_{x=0}^{w-1} \sum_{y=0}^{h-1} f(x, y) \cdot e^{-2\pi i(k_x x/w + k_y y/h)} \tag{26}$$

where $k_x$ and $k_y$ correspond to the wavenumber components in the horizontal and vertical directions, respectively, and $i$ is the imaginary unit. The power at each wavenumber component $(k_x, k_y)$ is given by the square of the magnitude spectrum of $F(k_x, k_y)$, that is,

$$S(k_x, k_y) = |F(k_x, k_y)|^2 = \text{Re}[F(k_x, k_y)]^2 + \text{Im}[F(k_x, k_y)]^2 \tag{27}$$

The scalar wavenumber is defined as:

$$k = \sqrt{k_x^2 + k_y^2} \tag{28}$$

which represents the magnitude of the spatial frequency vector, indicating how rapidly features change spatially regardless of direction. Then, the energy distribution at a spatial frequency corresponding to k is defined as:

$$S(k) = \sum_{(k_x, k_y):\sqrt{k_x^2 + k_y^2}=k} S(k_x, k_y) \tag{29}$$

Given the spatial energy frequency distribution for observations $S(k)$ and predictions $\hat{S}(k)$ , we perform normalization over $\mathbf{K}$, the set of wavenumbers, as defined in Equation 30. This is to ensure that the sum of the component sums up to 1 which exhibits pdf-like property.

$$S(k) \leftarrow \frac{S(k)}{\sum_{k \in \mathbf{K}} S(k)}, \quad \hat{S}(k) \leftarrow \frac{\hat{S}(k)}{\sum_{k \in \mathbf{K}} \hat{S}(k)} \tag{30}$$

Finally, the SpecDiv is formalized as follows:

$$\mathcal{M}_{SpecDiv} = \sum_{k} S(k) \cdot \log(S(k)/\hat{S}(k)) \tag{31}$$

where $S(k), \hat{S}(k)$ are the power spectra of the target and forecast along space continuum.

## C  EXPERIMENTAL DETAILS

### C.1  KOLMOGOROV FLOW

**Model training and inference**. All models are trained over 256 epochs, optimized with ADAMW Loshchilov & Hutter (2017), with a batch size of 64, learning rate of $2 \times 10^{-4}$, and a weight decay of $1 \times 10^{-3}$. During diffusion inference, we apply 64 denoising steps with 1-step LMC correction, $\gamma = 10^{-2}$, and $\tau = 3e^{-2}$. Training and inference are performed using 1x A100 NVIDIA GPU in a 100GB memory node.

**Ablation**. In order to build the best ensembles based on MC-dropout and IC perturbation strategy, we perform the following ablation: (i) vary the probability of dropout during inference (MC-dropout), (ii) introduce Gaussian noise to initial condition (IC) following $\varepsilon \sim \mathcal{N}(0, f\mathbf{I})$, where $f \in [0, 1]$. Figure 13 demonstrates that a dropout rate $p = 0.1$ and perturbation factor $f = 0.1$ yield the best ensembles.

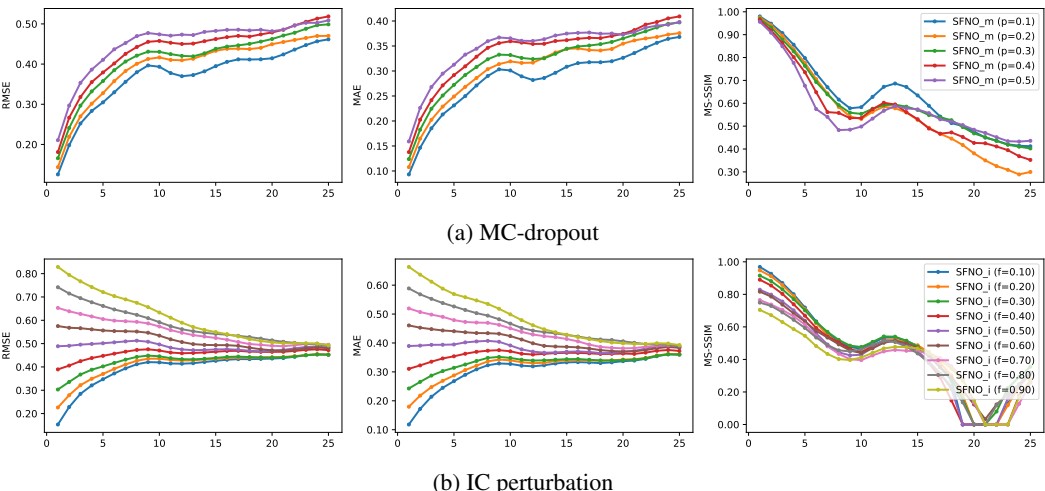

(a) MC-dropout

(b) IC perturbation

Figure 13: Ablating the best strategies to yield the optimal ensembles for MC-dropout and IC perturbation for Kolmogorov Flow, by varying dropout probability $p$ and Gaussian noise factor $f$ respectively.

### C.2  SHALLOW WATER EQUATION

We define a new coordinate system on the spherical domain $\mathbf{x} \in \mathcal{S}^2$ in terms of longitude $\varphi \in [0, 2\pi]$ and colatitude $\theta \in [0, \pi]$ Bonev et al. (2023). The unit vector $\mathbf{x}$ can then be reparamterized as $(\cos(\varphi)\sin(\theta), \sin(\varphi)\sin(\theta), \cos(\theta))^T$. Given this coordinate transform, we define the set of differential equations describing SWE:

$$\partial_t \varphi + \nabla \cdot (\varphi u) = 0$$
$$\partial_t (\varphi u) + \nabla \cdot T = \mathcal{S} \tag{32}$$

with initial conditions $\varphi = \varphi_0, u = u_0$. The state vector $(\varphi, \varphi u^T)^T$ includes both the geopotential layer depth $\varphi$ (representing mass) and the tangential momentum vector $\varphi u$ (indicative of discharge). Within curvilinear coordinates, the flux tensor $T$ can be expressed using the outer product $\varphi u \otimes u$. The right-hand side of the equation features flux-related terms, such as the Coriolis force.

**Experimental setup**. We use spectral method Giraldo (2001) to solve the PDE on an equiangular grid with a spatial resolution of $120 \times 240$ and 60-second timesteps. Time-stepping is performed using the third-order Adams-Bashford scheme and snapshots are taken every 5 hour for a total of 12 days, keeping the last 32 temporal sequences of vorticity outputs. The parameters of the PDE, such as gravity, radius of the sphere and angular velocity, are referenced to the Earth.

**Model training and inference**. All models are trained over 256 epochs, optimized with ADAMW Loshchilov & Hutter (2017), with a batch size of 64, learning rate of $2 \times 10^{-4}$, and a weight decay of $1 \times 10^{-3}$. During diffusion inference, we apply 1024 denoising steps with 1-step LMC correction, $\gamma = 10^{-2}$, and $\tau = 3e^{-2}$. Training and inference are performed using 1x A100 NVIDIA GPU in a 100GB memory node.

**Ablation**. In order to build the best ensembles based on MC-dropout and IC perturbation strategy, we perform the following ablation: (i) vary the probability of dropout during inference (MC-dropout), (ii) introduce Gaussian noise to initial condition (IC) following $\varepsilon \sim \mathcal{N}(0, f\mathbf{I})$, where $f \in [0, 1]$. Figure 14 demonstrates that a dropout rate $p = 0.2$ and perturbation factor $f = 0.1$ yield the best ensembles.

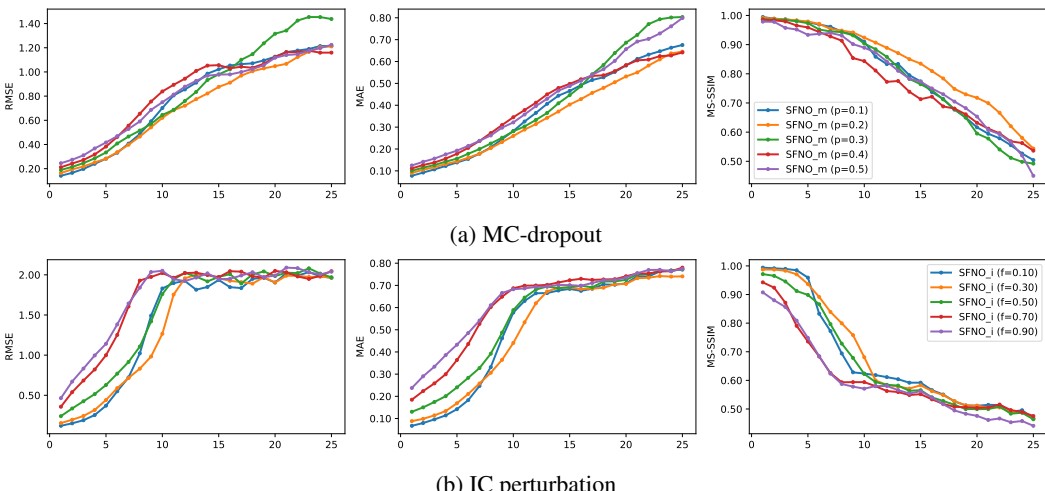

Figure 14: Ablating the best strategies to yield the optimal ensembles for MC-dropout and IC perturbation for Shallow Water Equation, by varying dropout probability $p$ and Gaussian noise factor $f$ respectively.