# OpenReview forum: "Cohesion: Coherence-Based Diffusion for Long-Range Dynamics Forecasting"
_ICLR.cc/2025/Conference — Submitted to ICLR 2025_

### Official Review · Reviewer_2JUM · 2024-10-23

**Soundness:** 2
**Presentation:** 2
**Contribution:** 1
**Rating:** 3
**Confidence:** 3

**Summary:**

The paper proposes Cohesion, a model that combines a deterministic latent autoregressive component and a diffusion model for probabilistic dynamics forecasting. The high-level intuition of the model is as follows: first the current state is encoded to a compressed latent space. Then, one or more steps of a deterministic autoregressive model are applied, after which the predictions in the latent space are decoded back to the data space to get the initial prediction(s). In terms of the Reynold decomposition, the paper interprets this initial prediction as the coherent component of the flow. To resolve the fluctuating component, a diffusion model, which is conditioned on the predicted coherent component, is used to ‘stochastically refine’ the states. Cohesion is evaluated against the Spherical Fourier Neural Operator (SFNO) on Kolmogorov flow and Shallow Water Equation benchmarks, in terms of point wise-metrics like MSE, as well as structure-based and physics-based metrics.

**Strengths:**

**S1:** The model is evaluated in terms of spectral divergence, a measure of divergence between the energy distribution at different frequencies. Indeed, for chaotic systems that can have strongly diverging trajectories for even slightly different initial conditions, point-wise metrics become meaningless for long simulation horizons, and the statistical properties of the system should be investigated.

**S2:** The interpretation of the deep Koopman operator as predicting the coherent part of the flow and of the diffusion model as refining the fluctuating component is intuitive and provides a physics-inspired motivation for Cohesion.

**S3:** The method supports different sampling strategies, dubbed trajectory planning and autoregressive. Autoregressive shows higher quality results, but requires more computational effort, whereas trajectory planning is more computationally efficient due to denoising the entire trajectory at once. As such, the sampling strategies provide an intuitive way to trade computational budget for accuracy.

**Weaknesses:**

**W1:** I found the presentation of the paper at some parts counterintuitive and confusing. In particular, Section 3 (the method section) starts with a quite extensive explanation of score-based generative models and zero-shot super resolution, while this concerns background material introduced in other works. I think the paper would benefit from merging this part with the text around Eq. 3 in the background section. This would make it easier for the reader to distinguish already established methods/algorithms and the novel aspects introduced in this paper.

**W2:** The experimental evaluation has two weak aspects:

* The method is compared against a single baseline only, the SFNO. There are other popular probabilistic methods that utilize a predictor-refinement approach, for example those that are cited in Section 2, e.g. Lippe et al. 2024; Srivastava et al. (2023); Yu et al. (2023); Mardani et al. (2024). It would greatly improve the paper to compare against a selection of those methods, and explain how the key conceptual differences in your approach relative to those works lead to different results. In addition, the work of Bergamin et al. [1] is relevant since it also considers two sampling modes that are highly similar to trajectory planning and autoregressive forecasting as described in the paper.
* The SFNO is not a suitable model for the Kolmogorov flow experiment, since the geometry in this experiment is not spherical. Do you have a specific reason to not compare to the regular FNO instead here (in addition to other baselines that would be good to add)?

**W3:** After reading the paper, it remains unclear to me what the key novel insights are relative to prior papers. Diffusion-based refinement techniques have already been proposed, as cited in Section 2 of this work. In addition, the zero-shot conditioning on noisy or partial observations in the context of PDEs has already been established in [2]. As I currently understand, the novelty here lies primarily in the architectural choice of using the encoder-operator-decoder module to get the initial prediction. In itself, this seems a conceptually small change relative to earlier frameworks that show that similar predictor-refinement approaches work well. If using this architecture in the prediction-refinement setting could lead to a substantial performance increase or otherwise interesting results, these could be novel insights, but this is not investigated in the paper. I think it would help to get your point across if you contrast your work against the most similar related papers more explicitly, and highlight the differences between them.

**References:**

[1] Bergamin et al., 2024. Guided Autoregressive Diffusion Models with Applications to PDE Simulation. https://openreview.net/forum?id=1avNKFEIOL

[2] Rozet & Louppe, 2023. Score based data assimilation. https://arxiv.org/abs/2306.10574

**Questions:**

**Q1:** Did you investigate whether the predicted conditioning prior actually aligns with the coherent part of the flow? I.e., does it predict some kind of average (or perhaps localized/moving average) behavior of the dynamics in space and/or time?

**Q2:** It is unclear to me how the metrics are calculated over multiple samples from the probabilistic models. Did you simply average the metrics over those samples, or take a best-of-K like approach? What is the effect of the stochastic sampling on the calculation of those metrics using your approach?

**Q3:** Please add labels on the horizontal axes of Figures 5 and 7.

**Q4:** How should Figure 6 be interpreted? Is the spherical geometry here somehow projected on the 2D image? Is there a reason why the signal is only present at the top half of the image?

**Q5:** Can you also provide plots of the spectral divergence over time, of Cohesion, the baseline methods, and coherent-only?

**Q6:** Please comment on W2 and W3.

---

> ### Author Response · Authors · 2024-11-23
>
> We would like to thank Reviewer 2JUM for their constructive feedback! Please find our response below:
>
> > On novelty (W1/W3)
>
> We moved several background sections e.g., zero-shot forecasting and score-based diffusion to the appendix and reframe them as background. We then attempt to make our contributions clearer, with more supporting evidence accompanying this rebuttal:
>
> - We propose __formal connection__ between conditional diffusion and Reynold's decomposition framework, and unify existing works (on diffusion-based PDE solvers) based upon this framework.
> - This unifying framework then allows us to demonstrate the __sufficiency of low-order, linearized__ conditioning prior for stable and skillful long-range forecast of nonlinear dynamics.
> - We strengthen our claims by studying the __scaling properties__ of diffusion-based PDE solvers in a manner that is closer in concept to fluid dynamics.
>
> > On baselines (W2)
>
> We have added probabilistic formulation of UFNO and the classical (tensorized) FNO, all with identical parameter budget as our Cohesion framework. These models are implemented off-the-shelf from https://github.com/neuraloperator/neuraloperator. The table below summarizes the results after evaluating for the best ensembling strategies (IC-perturb / MC-dropout / Multi-checkpoints), across Kolmogorov/SWE experiments, at the final timestep T.
>
> |                 | **Cohesion (R=1)** | **Cohesion (R=T)** | **TFNO (ensemble)** | **UFNO (ensemble)** | **SFNO (ensemble)** |
> |-----------------|--------------------|--------------------|---------------------|---------------------|---------------------|
> | **RMSE (↓)**    | **0.31 / 0.25**    | **0.37 / 0.40**    | 0.40 / 1.52         | 0.42 / 0.92         | 0.49 / 0.93         |
> | **MAE (↓)**     | **0.25 / 0.15**    | **0.27 / 0.20**    | 0.30 / 1.32         | 0.35 / 0.51         | 0.40 / 0.52         |
> | **MS-SSIM (↑)** | **0.68 / 0.95**    | **0.45 / 0.90**    | 0.01 / 0.01         | 0.21 / 0.42         | 0.32 / 0.43         |
>
> > On more ablation (W3)
>
> We argue, using the unifying conditional diffusion -- Reynolds decomposition framework, that existing works like Lippe et al., (2024), Srivastava et al. (2023) is conceptually similar to our Cohesion framework in autoregressive mode (R=1) at which correction is performed, conditional on previous forecast. In this case, we find substantial increase in inference speed, if we perform inference as trajectory planning (R=T), with minimal deterioration in skill. In a sense, our framework provides a __generalization__ to how forecasting is performed with e.g., Lippe et al., as a special case of R=1, video generation task as R=T, and anywhere in between as R=[1..T], for a more flexible design and provides strategies (temporal convolution / Markov blanket) to ensure consistency.
>
> > On Q1
>
> Indeed, from our comparison of spectral profiles (figure in the paper) that compare between the coherent flow and post-correction, the former is just capturing the low-frequency, low-mode variability signal. Upon further inspection of the structure (qualitative), it is akin to compression / applying a low-pass Filter.
>
> > On Q2
>
> For the metrics, we compute the average over all testing samples and across 5-member ensembles generated randomly.
>
> > On Q4
>
> This SWE is forced with Galewsky initial condition that models mid-latitude instability, and hence only the northern hemisphere has signal.

---

> ### Author Response · Authors · 2024-11-23
>
> > More on scaling properties
>
> Since the purpose of this work is primarily to probe the limits of diffusion-based PDE solver, we conducted two more ablations:
>
> __(1) Choice of mean flow estimator__: We substitute deep Koopman operator, that both truncates and linearizes the PDE solution to mimic mean flow in Reynold's decomposition, with xNOs, and find substantial degradation at long-range rollouts.
> |                 | **Cohesion (TFNO)** | **Cohesion (UFNO)** | **Cohesion (SFNO)** | **Cohesion (ROM)** |
> |-----------------|---------------------|---------------------|---------------------|--------------------------|
> | **RMSE (↓)**    | 0.41 / >2.00        | 0.42 / >2.00        | 0.46 / 1.51         | **0.37 / 0.40**          |
> | **MAE (↓)**     | 0.30 / >2.00        | 0.31 / >2.00        | 0.36 / 0.65         | **0.27 / 0.20**          |
> | **MS-SSIM (↑)** | 0.01 / 0.30         | 0.27 / 0.33         | 0.39 / 0.50         | **0.45 / 0.90**          |
>
> __Conclusion 2: ROMs stabilize the reverse conditional denoising process, while remaining robust and skillful__
>
> __(2) Scaling property of ROM__: We ablate the ROM with orders-of-magnitude reduction in parameter size. Even with 2 orders magnitude smaller ROM, Cohesion is competitive even with the best probabilistic version of xNOs.
>
> |                 | **Cohesion (x$10^{-2}$)** | **Cohesion (x$10^{-1}$)** | **Cohesion (x$10^0$)** | **Reference (best xNO)** |
> |-----------------|---------------------------|---------------------------|------------------------|--------------------------|
> | **RMSE (↓)**    | 0.39 / 0.95               | 0.38 / 0.90               | 0.37 / 0.40            | 0.40 / 0.92              |
> | **MAE (↓)**     | 0.30 / 0.50               | 0.28 / 0.49               | 0.27 / 0.20            | 0.30 / 0.51              |
> | **MS-SSIM (↑)** | 0.27 / 0.59               | 0.40 / 0.60               | 0.45 / 0.90            | 0.32 / 0.43              |
>
> __Conclusion 3: Diffusion scales with coarse flow approximation__
>
> In summary, we do not intend to propose another diffusion-based PDE solver, but rather to provide a __unifying framework__ that brings diffusion closer to the language of fluid dynamics, and how ideas in the latter (e.g., mean / stochastic flow) can be seamlessly incorporated to explore and probe the scaling limits of diffusion as a promising tool for long-range dynamics solver.

---

> > ### Comment · Reviewer_2JUM · 2024-11-25
> > **Thank you for the rebuttal**
> >
> > I thank the authors for their extensive rebuttal. Please find my response below:
> >
> > **Regarding W1/W3 - novelty and contributions:** I appreciate that the authors summarize their key contributions in the rebuttal, and propose to move part of section 3 to the appendix or background. However, editing the manuscript to reflect the proposed changes requires almost completely rewriting section 3. Without seeing the updated paper, it is impossible to assess the updated manuscript.
> >
> > **Regarding W2 - baselines:**  I appreciate the newly added baselines. The results seem promising. Can you add the plots similar to Figures 4 and 5 to the paper or the appendix so that the new results can be examined in more detail? Further, the first point under W2 is unaddressed.  If you are claiming that your method generalizes e.g. Lippe et al., it would be good to add PDE-Refiner (or another predictor-refiner-like method) as a baseline.
> >
> > **Regarding choice of mean flow estimator:** These results look promising. Can you also explain in more detail _why_ the xNO models perform worse than your ROM as mean flow predictors?
> >
> > Overall, I think the authors' rebuttal goes in the right direction to more clearly reflect the contributions (W1/W3) and provides substantially improved experimental results of ablations and baselines (W2). Still, incorporating this into the manuscript requires substantial rewriting of sections 3 and 4. Without seeing the updated paper, it is difficult to judge the quality, and assessing an updated version (if/when available) might require another full review, given the extent of the changes. Additionally, I would encourage the authors to add at least one predictor-refiner method (e.g. Lippe et al.) as baseline.

---

> > > ### Author Response · Authors · 2024-11-25
> > >
> > > We appreciate Reviewer 2JUM for the constructive feedback and encouragement!
> > >
> > > - We do agree that in light of the revision requests and the near-end timeline of the discussion period, non-trivial changes to the manuscript is warranted (e.g., reframing of multiple sections, adding another baselines). We are currently planning to conduct these additional experiments and report their results in this forum for the community to view.
> > >
> > > - With respect to the baselines, we show that diffusion is robust even in low-order, highly compressed prior, as demonstrated by orders-of-magnitude smaller ROMs. One of the reasons for the failure of xNOs, we postulate, is due to the lack of truncation, where high-frequency signals are retained and propagated over long-rollouts, causing instabilities. But this is merely speculation, and one would need to systematically ablate the stability w.r.t. the number of modes retained, a question for future work.
> > >
> > > With all that said, we do want to iterate that our work's first and foremost objective is to provide a dynamics-based perspective / formalism to navigate and analyze many works on diffusion-based PDE solvers, though often spoken with different jargons e.g.,:
> > >
> > > - __mean flow__: first guess, control run, etc
> > > - __stochastic variation__: correction, post-processing, etc
> > > - __auxiliary contexts__: physics constraints, statistics, etc.
> > >
> > > Though the specific instantiation of these components differ and often entangled, we argue that the core ideas follow closely with classical fluid dynamics, hence our effort to formally draw the connection and probe the limits with a very simplified method (e.g., ROM). We do understand that with any framework, it can never fully capture the entire spectrum of works, but we hope that it provides some clarity and intuition to navigate the fast-growing literature.

---

### Official Review · Reviewer_yE9B · 2024-10-26

**Soundness:** 3
**Presentation:** 2
**Contribution:** 3
**Rating:** 5
**Confidence:** 4

**Summary:**

This paper presents Cohesion, a Coherence-Based Diffusion Model for Long-Range Dynamics Forecasting. The model leverages the concept of Reynolds Averaged Navier-Stokes (RANS) as conditioning priors to address two key issues in diffusion-based autoregressive models: (1) instability in long-term predictions, and (2) the computational inefficiency of generating priors. The authors utilise a Koopman-based reduced order model to efficiently generate these priors, thereby speeding up the forecasting process. Standard fluid dynamics principles and diffusion models are adapted to support this framework. Furthermore, cohesion acts as a refinement mechanism that aggregates temporal sequences from the model’s output, improving performance. The proposed approach is evaluated on two benchmark fluid systems. While the work is well-motivated and promising, there are critical issues with the presentation and formulation (detailed below).

**Strengths:**

The targeted problems are both interesting and critical for applying diffusion models to autoregressive forecasting. The main ideas and design choices presented in this paper are well-motivated and promising.

**Weaknesses:**

### Presentation
1.	The authors claim that trajectory planning, a concept central to reinforcement learning (RL), is crucial to Cohesion. However, they do not adequately explain the operational or conceptual connections. Although they mention reframing forecasting as trajectory planning, no clear relationship to RL principles—beyond standard autoregressive or multi-step predictions—is evident. Without a specific RL objective formulation or a direct link to decision-making scenarios, the reference to RL seems forced and mislead readers.

2.	The paper's use of terminology is potentially confusing and could be considered an abuse of terms. Specifically, the names "Cohesion" and "coherent" are too similar, which creates ambiguity about their distinct roles. From personal understanding, the cohesion refers to the whole framework whereas “coherent” describes a predictable component of the dynamical system. While, as the figures 1,2,3 and experiment sections, the "cohesion " seems refer to the temporal-aggregation component. In addition, given the two words have specific meanings in the scientific community, the author should be careful and avoid potential confusions and abusing of terminology.

3. Koopman Operator Introduction (Lines 253-256): The introduction of the Koopman operator is vague and contains some mathematical issues: (1) the text does not discuss that practical implementations use a finite-dimensional approximation. (2) The domain and mapping of the encoder and decoder are not clearly defined. (3) No clear definition for $\mathcal{G}$ and $\mathcal{G}_E(\mathcal{X})$.

4. The numerical results are discussed too briefly, with only three lines dedicated to each experiment (Lines 378-380, 432-434). More in-depth discussion and analysis are needed to properly interpret the findings.

5. There is only one baseline methods. Additional baselines, particularly diffusion-based models, should be included for a more comprehensive comparison.

## Others
1.	The paper uses incorrect citation formatting. Citations should be enclosed in parentheses, e.g., "(xxx et al., year)," when they are not acting as the subject or object within sentences.
2.	Figures 5, 7, 8, and 9 lack axis descriptions
3.	Several acronyms, such as Number of Function Evaluations (NFE), are introduced without definition at first mention. All abbreviations should be defined before initial use to ensure clarity.

**Questions:**

1. The Spherical Fourier Neural Operator (SFNO) is used as the only baseline despite being designed for spherical domains, whereas both demonstrated cases in the paper use standard square domains. This choice makes SFNO an unsuitable and potentially misleading baseline. It is unclear why the authors included this baseline, as it does not align with the problem setting.

2. The use of Reynolds decomposition is indeed a central aspect of the paper, and this core idea is not clearly stated. While RANS typically deals with time-averaged components, the authors extend it to a time-dependent setting without discussion on this extension and theoretical foundation. It remains unclear how the authors achieved this extension and ensured its validity.

3. The authors claim improved inference efficiency through incorporating Koopman-based ROM; however, no comparison with other baselines is provided to support this claim.

---

> ### Author Response · Authors · 2024-11-23
>
> We would like to thank Reviewer yE9B for their constructive feedback! Please find our response below:
>
> > Connection with RL principles
>
> We agree that we draw inspiration from the Diffuser paper to solve long-range forecasting of nonlinear dynamical system. Although not explicitly demonstrated in the paper, we clarify in the future work section that additional reward function could easily be incorporated using Diffuser's framework, such as the use of physics-based constraints for stability guidance e.g., [1].
>
> > On clarity
>
> We attempted to make the notation and terms defined early on in the paper to avoid confusion. For instance, a framework such as _Cohesion_ will be italicized.
>
> > On baselines
>
> We have added probabilistic formulation of UFNO and the classical (tensorized) FNO, all with identical parameter budget as our Cohesion framework. These models are implemented off-the-shelf from https://github.com/neuraloperator/neuraloperator. The table below summarizes the results after evaluating for the best ensembling strategies (IC-perturb / MC-dropout / Multi-checkpoints), across Kolmogorov/SWE experiments, at the final timestep T.
>
> |                 | **Cohesion (R=1)** | **Cohesion (R=T)** | **TFNO (ensemble)** | **UFNO (ensemble)** | **SFNO (ensemble)** |
> |-----------------|--------------------|--------------------|---------------------|---------------------|---------------------|
> | **RMSE (↓)**    | **0.31 / 0.25**    | **0.37 / 0.40**    | 0.40 / 1.52         | 0.42 / 0.92         | 0.49 / 0.93         |
> | **MAE (↓)**     | **0.25 / 0.15**    | **0.27 / 0.20**    | 0.30 / 1.32         | 0.35 / 0.51         | 0.40 / 0.52         |
> | **MS-SSIM (↑)** | **0.68 / 0.95**    | **0.45 / 0.90**    | 0.01 / 0.01         | 0.21 / 0.42         | 0.32 / 0.43         |
>
> __Conclusion 1: Cohesion outperforms all xNOs baselines__
>
> > On ROM ablation
>
> We substitute deep Koopman operator, that both truncates and linearizes the PDE solution to mimic mean flow in Reynold's decomposition, with xNOs, and find substantial degradation at long-range rollouts.
> |                 | **Cohesion (TFNO)** | **Cohesion (UFNO)** | **Cohesion (SFNO)** | **Cohesion (ROM)** |
> |-----------------|---------------------|---------------------|---------------------|--------------------------|
> | **RMSE (↓)**    | 0.41 / >2.00        | 0.42 / >2.00        | 0.46 / 1.51         | **0.37 / 0.40**          |
> | **MAE (↓)**     | 0.30 / >2.00        | 0.31 / >2.00        | 0.36 / 0.65         | **0.27 / 0.20**          |
> | **MS-SSIM (↑)** | 0.01 / 0.30         | 0.27 / 0.33         | 0.39 / 0.50         | **0.45 / 0.90**          |
>
> __Conclusion 2: ROMs stabilize the reverse conditional denoising process, while remaining robust and skillful__
>
> In summary, in terms of __scaling limit__, we argue that a lightweight, interpretable conditioning factor to capture mean flow is __sufficient__ for skillful diffusion-based PDE solver, and highlights the latter's promise of revolutionizing (if not already) the field.
>
> __References:__
>
> [1] Schiff, Yair, et al. "DySLIM: Dynamics Stable Learning by Invariant Measure for Chaotic Systems." arXiv preprint arXiv:2402.04467 (2024).

---

> ### Comment · Reviewer_yE9B · 2024-11-24
>
> Thank you for your detailed response and the additional clarifications. I appreciate the effort in addressing the feedback, particularly the inclusion of new numerical results. Below are my thoughts on the review and some remaining concerns:
>
> - **Connection to RL:** Thanks for the clarification of RL principle and the provided reference, which is a good idea to add such constraint. However, from reviewer’s view, this is still far from RL principle, where decision-making/control are central objectives. Limiting RL principles to the context of the diffuser policy seems overly narrow and could be misleading. Additionally, the explanation provided appears more aligned with future work rather than the scope of this paper. I suggest revising the RL-related sections with a broader and more accurate framing.
>
> - **Clarity:** Thank you for the clarification; it is now clearer to me. I encourage the authors to further refine the manuscript to ensure that the key concepts are presented clearly and are easy to follow.
>
> - **Additional Baselines:** The inclusion of additional baselines is appreciated and partially addresses my concerns. However, as I stated in Weakness 5, comparisons with diffusion-based models are still missing. Adding such models would make the evaluation more comprehensive, e.g., Rühling, et al., (2024) and Gao, et al., (2024).
>
> - **Ablation Study:** I am glad to see the provided ablation study, which is a valuable addition and demonstrative.
>
> - **Unresolved Points:** Weakness 3 and Question 2 remain unaddressed.
>
> In conclusion, based on the above unsolved concerns, I maintain my score but look forward to further discussions, particularly if the RL-related sections can be adjusted in the updated manuscript.
>
>
> - References
>
>   - Rühling Cachay, Salva, et al. "Dyffusion: A dynamics-informed diffusion model for spatiotemporal forecasting." Advances in Neural Information Processing Systems 36 (2024).
>
>   - Gao, Han, Sebastian Kaltenbach, and Petros Koumoutsakos. "Generative learning for forecasting the dynamics of high-dimensional complex systems." Nature Communications 15.1 (2024): 8904.

---

> > ### Author Response · Authors · 2024-11-24
> >
> > We appreciate Reviewer yE9B for their constructive feedback!
> >
> > > Unresolved Points: Weakness 3 and Question 2 remain unaddressed.
> >
> > - __On time-dependent Reynold's decomposition__
> >
> > Indeed, we assume a time-dependent Reynolds decomposition, which departs from the traditional time-averaged approach used in RANS. In our framework, the mean component $\( \bar{u}(t) \)$ is defined as a local temporal average (or smoothed representation), while the fluctuating component $\( u'(t) \)$ captures deviations over shorter time scales. This decomposition allows us to model both instantaneous dynamics and the interplay between sub-sequences.
> >
> > We frame the problem in terms of three cases:
> >
> > 1. **Autoregressive (R=1)**: The __Markovian assumption__ ensures time independence, aligning with traditional Reynolds decomposition where fluctuations depend only on the immediate state.
> >
> > 2. **Trajectory Planning (R=T)**: Here, the reverse denoising process maximizes the log-likelihood over the __entire trajectory__, implicitly capturing dependencies through the mean-fluctuation framework.
> >
> > 3. **Intermediate Cases (R > 1, R < T)**: For these cases, we leverage the __pseudo-Markov blanket__ theorems proposed by Rozet et al. (2023) [1]. Their results show that it is possible to isolate dependencies within a sub-sequence such that the log-likelihood maximization is locally Markovian with respect to the next sub-sequence. This ensures that the Reynolds decomposition remains valid within each sub-sequence (similar argument as (2) but within sub-sequence), even in a time-dependent setting.
> >
> > > Koopman Operator
> >
> > Indeed, though in theory, the Koopman operator is able to linearize any nonlinear system in __infinite__ dimensionality, in practice this might be difficult and tricky. In cases where the invariant measures (e.g., attractors) of the mean flow modeled are high-dimensional, the Koopman operator has to scale. However, we argue that the benefit of such linearization is not to fully capture the potentially high-dimensional mean flow, but to provide other interpretable benefits, such as linear stability analysis. Nonetheless, the addition of nonlinear diffusion module can assist in resolving higher frequency signal and small-scale physics, and mitigate such difficulties.
> >
> > We clarified the definition of $\{\mathcal{G}_E, \mathcal{G}_D\} \in \mathcal{G}$ as the invertible models between the state and observables in canonical Koopman terms (for our case is a simple convolution-based pair of encoder $\mathcal{G}_E$ and decoder $\mathcal{G}_D$), where the latent space where the Koopman operator acts has reduced dimensionality relative to the physical-data space i.e., $n_d << n_x$.
> >
> > We will clarify all these in the manuscript.
> >
> > > Additional Baselines
> >
> > We do not intend to propose another diffusion-based PDE solver, but rather to provide a __unifying framework__ that threads together existing works, and brings diffusion closer to the language of fluid dynamics, and how ideas in the latter (e.g., mean / stochastic flow) can be seamlessly incorporated to explore and probe the scaling limits of diffusion as a promising tool for long-range dynamics solver. For instance, in referenced works like [2] and [3], we argue that the former is an instantiation of Cohesion in autoregressive mode (R=1) at which correction is performed, conditional on previous forecast, and the latter is a great example of Cohesion in trajectory planning mode (R=T). Both works only differ in how they incorporate additional conditioning context (e.g., physics, long-range statistics). In a sense, our framework provides a __generalization__ to how forecasting is performed, from two extreme cases, and anywhere in between as R=[1..T], for a more flexible design and provides strategies (e.g., model-free temporal convolution) to ensure consistency.
> >
> > __References__:
> >
> > [1] Rozet, F., & Louppe, G. (2023). Score-based Data Assimilation. ArXiv, abs/2306.10574.
> >
> > [2] Rühling Cachay, Salva, et al. "Dyffusion: A dynamics-informed diffusion model for spatiotemporal forecasting." Advances in Neural Information Processing Systems 36 (2024).
> >
> > [3] Gao, Han, Sebastian Kaltenbach, and Petros Koumoutsakos. "Generative learning for forecasting the dynamics of high-dimensional complex systems." Nature Communications 15.1 (2024): 8904.

---

> > > ### Author Response · Authors · 2024-11-24
> > >
> > > > Connection to RL
> > >
> > > We would like to clarify that our paper is not intended to apply RL-based diffusion framework in fluid dynamics, but rather to give credits as to where our inspiration comes from. With that said, we find several connections that might be worth exploring for future work, e.g., in the policy-based RL, one attempts to variationally maximize the log policy gradient i.e., $\mathbb{E} \left[ \sum_{t=0}^T \nabla_{\theta} \log \pi_\theta(a_t | s_t) R(\tau) \right]$, where the action $a_t$ and $s_t$ can be thought of as the state and conditioning prior (e.g., mean flow). This objective is similar to the guidance performed during the reverse denoising process. The missing part here would be the incorporation of a reward function $R(\tau)$ given a specific policy function $\tau \sim \pi_\theta$. This reward can be designed flexibly e.g., to capture long-term statistics, some conservation constraints, etc. Similar formulation can be extended to value-based methods where the objective is to maximize the reward function $R$ instead.
> > >
> > > This RL formulation could indeed be interesting for many fluid dynamics applications, such as in simulation, control, hybrid integration with numerical solvers, and even in operational setting where data assimilation of sparse observations is performed such that the estimated state should best reproduce them (as additional guidance / reward).

---

> > > > ### Comment · Reviewer_yE9B · 2024-11-26
> > > >
> > > > I appreciate the authors' further responses.
> > > >
> > > > However, I still have questions regarding the RL formulation. From reviewer's perspective, the role of the RL formulation in this work appears to overlap with the score function, which can be interpreted similarly to the value function in RL as noted by the authors. Why is it necessary to explain the same concept from both perspectives? For instance, Rozet et al. (2023) applied the score-based diffusion model for data assimilation using a single core concept without mentioning RL.
> > > >
> > > > I also appreciate the explanation of the missing $R(\tau)$ designed for different scenarios, but I believe it is not directly relevant to this work.
> > > >
> > > > Rozet, François, and Gilles Louppe. "Score-based data assimilation." Advances in Neural Information Processing Systems 36 (2023): 40521-40541.

---

### Official Review · Reviewer_HSiU · 2024-10-28

**Soundness:** 2
**Presentation:** 1
**Contribution:** 2
**Rating:** 3
**Confidence:** 4

**Summary:**

The paper proposes a diffusion-based approach for forecasting with dynamical systems that is able to generate the entire sequence in one conditional denoising pass. This is achieved by leveraging reconstruction guidance, where the conditioning information is a sequence of priors (one for each state of the final trajectory), generated by iteratively applying a (lightweight) reduced-order method (ROM) to the initial condition. The score network operates over subsequences, and temporal coherency is assured by applying temporal convolution with a small receptive window. The experiments are performed on two chaotic systems (Kolmogorov flow and Shallow Water), and the performance of the model as a probabilistic emulator is tested in terms of pixel-based metrics (RMSE, MAE), structure-based metrics (MS-SSIM), and physics-based metrics.

**Strengths:**

1. **Relevant topic.** The problem addressed in this paper (probabilistic emulation for PDEs) is an active area of research with important downstream applications, such as weather and climate modelling.
2. **Non-autoregressive approach.** The possibility of using a non-autoregressive sampling strategy from the diffusion model without compromising too much on accuracy is nice, and something of significance for the field of forecasting.
3. **Flexibie guidance with ROMs.** The use of reconstruction guidance is a very useful technique when the observation process changes over time, offering more flexibility as opposed to classifier-based approaches. Although the technique is not a contribution of this work, the paper is the first (as far as I am aware) to utilise as conditioning information a sequence of autoregressively-produced priors through a compute-efficient ROM. The experiment in which the authors condition on partially-observed fields of the ROM output highlights the flexibility of this technique and is relevant to real-life settings with partially-observed data.
5. **Wide range of metrics used for the experiments.**

**Weaknesses:**

1. **Unclear contributions/Lack of clear citations.** This approach shares striking similarities with the approach from Rozet et al. [1], but fails to properly reflect this throughout the main text. There should be a much more clear distinction between the contributions of this paper and what is taken from other works.
- The overall training and sampling from the score-based model relies heavily on the approach proposed by Rozet et al. [1], but this is not made clear in the paper. They also train the model on subsequences of length $W$ (justifying this approach from a Markov order perspective), and stitch these subsequences together at sampling time to generate arbitrary length trajectories in one go.
- The reconstruction guidance mechanism is exactly the same as the one proposed in Rozet et al. [1]. This is mentioned in the paper (L215), but it can be interpreted as if the authors propose a way to improve the numerical stability of the method, as opposed to using results from previous work.
- The similarity between the proposed framework and Rozet et al. [1] is reflected in the algorithms presented.
   - Algorithm 1 is the same as Algorithm 3 in Rozet et al. [1]
   - Algorithm 2 is the same as Algorithm 4 in Rozet et al. [1]
   - Algorithm 3 is the same as Algorithm 1 in Rozet et al. [1]
   - Algorithm 4 is the same as Algorithm 2 in Rozet et al. [1]. However, in the paper this is posed as a novel temporal convolution technique, rather than something already employed in Rozet et al. [1].

  I acknowledge that this paper adapts Rozet et al. [1]’s approach, making it suitable to other tasks (forecasting), as opposed to data assimilation. This is achieved by conditioning on those prior states, generated autoregressively through a ROM. This is a nice approach and equips Rozet’s method with the ability to perform forecasting, a task where their approach fails based on the observations from Shysheya et al. [2]. However, I do not think this is how the paper portrays the technique, and it is debatable whether this is enough of a contribution overall when considering the results (see below). At the very least, a clear paragraph on contributions should be included.
2. **Weak baseline for the empirical analysis.** Although the paper mentions that the SFNO approach [Bonev et al. [4]] is the state-of-the-art, I believe there are other probabilistic forecasting approaches that achieve stronger performance.
- Two works I have in mind are Lippe et al. [5] and Shysheya et al. [2], with the former being considered state-of-the-art. However, the main metric these works consider for forecasting is high correlation time, rather than the metrics used in this work. But based on the trajectories, they seem to maintain correlation with the ground truth for longer than Cohesion. It would be interesting to compute the high correlation time and compare it with some of these works, especially since the Kolmogorov dataset looks similar to the one in Shysheya et al. [2].
- As in Lippe et al. [5], I believe that another relevant (deterministic) baseline would be an MSE-trained UNet. In their experiments, it tends to achieve better results than FNO-based approaches.
3. **Lack of error bars in the results.** The performance plots lack error bars. Thus, it is hard to determine how significant the difference between methods is. This is especially the case between Cohesion (R=1) vs. Cohesion (R=T) (i.e. autoregressive vs. in one go), where having error bars is important to figure out whether the hit in performance by generating the entire trajectory in one go is significant.
4. **Copying from other papers without citing.** There are certain paragraphs/sections in the appendix which are directly taken from other works without specifying so. For example Appendix B.2. Structure-based metrics is the same as Appendix F.1.4. Multi-scale Structural Similarity Index Measure (MS-SSIM) in Nathaniel et al. [6], Appendix B.3 is very similar to Appendix F.2 in Nathaniel et al. [6]. It is ok to use the same definitions as in other works (in the end, the definitions of the metrics are what they are), but if the writing is so similar, I think you should at least cite the relevant work.

   Could you please review these sections and either rephrase them, or make it clear that they are heavily based on Nathaniel et al. [6]?

**Minor**

5. **Typos.** The paper contains several typos, I won’t include them all but examples are L091-spatiotempral, L125 - deterministic priors, L235 - should be $\nabla_{\mathbf{u}_k}$ , etc.
6. **Small labels in figures.** See for example y label in Figures 5, 7, legend in Figure 8.
7. **Unclear figures.** I appreciate the attempt to create a visual representation of the framework in Figure 1, but I find the figure confusing, without mentioning the colour coding of the bubbles, why there are two trajectories in b), etc.
8. **Lack of x label in Figure 5**. I believe that is the timestep $T$, but that should be labelled.
9. **Ablation hyperparameter choices.** In appendix C.1 you detail how you chose the dropout rate $p$ and perturbation factor $f$ for the baselines. However, those correspond to the lowest values you experimented with and it’s unclear whether going even lower would give better results or not. You’d like to obtain a convex function of your hyperparameters (i.e. worse performance for lower and higher values of the hyperparameter).

Overall, while the topic addressed by this paper is relevant, I believe it does not clearly differentiate its contributions to what already exists in the literature. For the empirical evidence, I believe the chosen baseline is not strong enough, and a more comprehensive comparison to other techniques is needed to thoroughly assess the effectiveness of the proposed method. Although in its current form the paper is incomplete, with proper baselines and a clear indication of novelty/contributions, it could represent a useful addition to the literature.

[1] Rozet, F., & Louppe, G. (2023). Score-based Data Assimilation. ArXiv, abs/2306.10574.

[2] Shysheya, A., Diaconu, C., Bergamin, F., Perdikaris, P., Hern'andez-Lobato, J.M., Turner, R.E., & Mathieu, E. (2024). On conditional diffusion models for PDE simulations.

[3] Qu, Y., Nathaniel, J., Li, S., & Gentine, P. (2024). Deep Generative Data Assimilation in Multimodal Setting. 2024 IEEE/CVF Conference on Computer Vision and Pattern Recognition Workshops (CVPRW), 449-459.

[4] Bonev, B., Kurth, T., Hundt, C., Pathak, J., Baust, M., Kashinath, K., & Anandkumar, A. (2023). Spherical Fourier Neural Operators: Learning Stable Dynamics on the Sphere. International Conference on Machine Learning.

[5] Lippe, P., Veeling, B.S., Perdikaris, P., Turner, R.E., & Brandstetter, J. (2023). PDE-Refiner: Achieving Accurate Long Rollouts with Neural PDE Solvers. ArXiv, abs/2308.05732.

[6] Nathaniel, J., Qu, Y., Nguyen, T., Yu, S., Busecke, J., Grover, A., & Gentine, P. (2024). ChaosBench: A Multi-Channel, Physics-Based Benchmark for Subseasonal-to-Seasonal Climate Prediction. ArXiv, abs/2402.00712.

**Questions:**

1. Could you please highlight the differences between this approach and Rozet et al. [1]? Is there anything that wasn’t captured in **W1**?
2. Could you also compute the high correlation time in the Kolmogorov (and SWE) experiment to compare with other results reported in the literature (i.e. Lippe et al. [5], Shysheya et al. [2])
3. While in the SWE experiment I understand the usefulness of the spherical embeddings used in SFNO, it is not clear to me why they would help in Kolmogorov, but I might have missed some relevant experimental setup detail that justifies it.
4. Could you provide error bars for your results?
5. If you provide comparisons to Lippe et al. [5] on the Kolmogorov flow experiment, could you also provide computational speed comparisons?
6. In L205 you assume a Gaussian observation process (as it is usually done). Would you be able to extend the framework to a non-Gaussian likelihood too?

---

> ### Author Response · Authors · 2024-11-23
>
> We would like to thank Reviewer HSiU for their constructive feedback. Please find our response specific to queries and concerns raised.
>
> > Lack of clear citations
>
> We moved several background sections e.g., zero-shot forecasting and score-based diffusion to the appendix and reframe them as background.  We have also, to the best of our ability, provide clear citations in places highlighted, and elsewhere.
>
> > Unclear contributions
>
> We attempt to make our contributions clearer, with more supporting evidence accompanying this rebuttal:
>
> - We propose __formal connection__ between conditional diffusion and Reynold's decomposition framework, and unify existing works (on diffusion-based PDE solvers) based upon this framework.
> - This unifying framework then allows us to demonstrate the __sufficiency of low-order, linearized__ conditioning prior for stable and skillful long-range forecast of nonlinear dynamics.
> - We strengthen our claims by studying the __scaling properties__ of diffusion-based PDE solvers in a manner that is closer in concept to fluid dynamics.
>
> > Lack of baselines
>
> We have added probabilistic formulation of UFNO and the classical (tensorized) FNO, all with identical parameter budget as our Cohesion framework. These models are implemented off-the-shelf from https://github.com/neuraloperator/neuraloperator. The table below summarizes the results after evaluating for the best ensembling strategies (IC-perturb / MC-dropout / Multi-checkpoints), across Kolmogorov/SWE experiments, at the final timestep T.
>
> |                 | **Cohesion (R=1)** | **Cohesion (R=T)** | **TFNO (ensemble)** | **UFNO (ensemble)** | **SFNO (ensemble)** |
> |-----------------|--------------------|--------------------|---------------------|---------------------|---------------------|
> | **RMSE (↓)**    | **0.31 / 0.25**    | **0.37 / 0.40**    | 0.40 / 1.52         | 0.42 / 0.92         | 0.49 / 0.93         |
> | **MAE (↓)**     | **0.25 / 0.15**    | **0.27 / 0.20**    | 0.30 / 1.32         | 0.35 / 0.51         | 0.40 / 0.52         |
> | **MS-SSIM (↑)** | **0.68 / 0.95**    | **0.45 / 0.90**    | 0.01 / 0.01         | 0.21 / 0.42         | 0.32 / 0.43         |
>
> __Conclusion 1: Cohesion outperforms all xNOs baselines__
>
> With regards to comparison with other autoregressive-based diffusion, like Lippe et al., [1] we argue that this is similar to our Cohesion framework in autoregressive mode (R=1) at which correction is performed, conditional on previous forecast. In this case, we find substantial increase in inference speed, if we perform inference as trajectory planning (R=T), with minimal deterioration in skill. In a sense, our framework provides a __generalization__ to how forecasting is performed with e.g., Lippe et al. [1], as a special case of R=1, video generation task as R=T, and anywhere in between as R=[1..T], for a more flexible design and provides strategies (temporal convolution / Markov blanket first proposed in the Diffuser paper [2] to solve RL) to ensure consistency.

---

> ### Author Response · Authors · 2024-11-23
>
> With respect to exploring the limits and design space of diffusion-based PDE solvers, we argue that a lightweight, low-order, linearized mean flow (inspired by Reynolds' decomposition), is sufficient, and that diffusion is _robust_ enough to resolve remaining residual flow. In order to illustrate this point and draw on the empirical connection with Reynold's decomposition, we perform two additional ablations: (1) substituting deep Koopman operator with TFNO/UFNO/SFNO, and (2) analyzing the scaling property of the ROM. Both analysis are performed on the trajectory mode (R=T) for Kolmogorov/SWE problems, at the final timestep T.
>
> __(1) Choice of mean flow estimator__: We substitute deep Koopman operator, that both truncates and linearizes the PDE solution to mimic mean flow in Reynold's decomposition, with xNOs, and find substantial degradation at long-range rollouts.
> |                 | **Cohesion (TFNO)** | **Cohesion (UFNO)** | **Cohesion (SFNO)** | **Cohesion (ROM)** |
> |-----------------|---------------------|---------------------|---------------------|--------------------------|
> | **RMSE (↓)**    | 0.41 / >2.00        | 0.42 / >2.00        | 0.46 / 1.51         | **0.37 / 0.40**          |
> | **MAE (↓)**     | 0.30 / >2.00        | 0.31 / >2.00        | 0.36 / 0.65         | **0.27 / 0.20**          |
> | **MS-SSIM (↑)** | 0.01 / 0.30         | 0.27 / 0.33         | 0.39 / 0.50         | **0.45 / 0.90**          |
>
> __Conclusion 2: ROMs stabilize the reverse conditional denoising process, while remaining robust and skillful__
>
> __(2) Scaling property of ROM__: We ablate the ROM with orders-of-magnitude reduction in parameter size. Even with 2 orders magnitude smaller ROM, Cohesion is competitive even with the best probabilistic version of xNOs.
>
> |                 | **Cohesion (x$10^{-2}$)** | **Cohesion (x$10^{-1}$)** | **Cohesion (x$10^0$)** | **Reference (best xNO)** |
> |-----------------|---------------------------|---------------------------|------------------------|--------------------------|
> | **RMSE (↓)**    | 0.39 / 0.95               | 0.38 / 0.90               | 0.37 / 0.40            | 0.40 / 0.92              |
> | **MAE (↓)**     | 0.30 / 0.50               | 0.28 / 0.49               | 0.27 / 0.20            | 0.30 / 0.51              |
> | **MS-SSIM (↑)** | 0.27 / 0.59               | 0.40 / 0.60               | 0.45 / 0.90            | 0.32 / 0.43              |
>
> __Conclusion 3: Diffusion is robust even with the coarsest approximation of mean flow__
>
> In summary, we do not intend to propose another diffusion-based PDE solver, but rather to provide a __unifying framework__ that threads together existing works, and brings diffusion closer to the language of fluid dynamics, and how ideas in the latter (e.g., mean / stochastic flow) can be seamlessly incorporated to explore and probe the scaling limits of diffusion as a promising tool for long-range dynamics solver.
>
> __References__
>
> [1] Lippe, Phillip, et al. "Pde-refiner: Achieving accurate long rollouts with neural pde solvers." Advances in Neural Information Processing Systems 36 (2024).
>
> [2] Janner, Michael, et al. "Planning with diffusion for flexible behavior synthesis." arXiv preprint arXiv:2205.09991 (2022).

---

> > ### Comment · Reviewer_HSiU · 2024-11-24
> >
> > Thank you for providing the additional results and clarifications regarding some of the addressed questions. However, I still have some concerns:
> >
> > *“We have also, to the best of our ability, provide clear citations in places highlighted, and elsewhere.”*
> > Would you be able to share the revised version of the manuscript? It would help to see how the changes you mentioned are reflected in the paper.
> >
> > **Contributions and comparison to SOTA**
> > The revised outline of the contributions is indeed clearer and more relevant than the current manuscript. However, to support the claim of providing *“stable and skillful long-range forecasts”*, I believe further empirical evidence is necessary—particularly in comparison to state-of-the-art methods.
> >
> > I am not saying that you necessarily need to outperform the state-of-the-art, but this comparison would help with situating your approach in the broader context of existing work. I believe that readers need to understand the relative strengths and limitations of your method, even if other methods incur a significantly higher computational cost (which would be part of the comparison).
> >
> > For instance, Figure 4 shows that your method starts to diverge from the ground truth at around $T = 19$. In comparison, other works seem to achieve longer adherence to the ground truth:
> > - Shysheya et al. [2]: Their method remains close to the ground truth for approximately 40 time steps (see Fig. 25 in their paper).
> > - PDE-Refiner [5]: In Fig. 13 of their work, samples maintain adherence to ground truth for up to 10 seconds, though it’s unclear how many time steps in your dataset correspond to 1 second (I suspect 1 time step is 0.2s?).
> >
> > Such comparisons would provide a clearer understanding of how your method performs relative to these baselines.
> >
> > **Extra baselines + lack of error bars** I appreciate the extra experimental evidence, and it is indeed useful to assess the comparison between using xNOs vs ROM to generate the conditioning information, and the scaling properties of the ROM.
> >
> > One weakness in this analysis is that **no error bars are provided**, so the actual (statistical) significance of the results is hard to assess. I have the same concern for the other results in the paper (also expressed as W3 in the rebuttal). Providing error bars is crucial for understanding the robustness and significance of your findings.
> >
> > **Generalisation of other frameworks** I think that the authors do not bring enough evidence to show how the Cohesion framework is a generalisation of other works in the literature such as PDE-Refiner or Dyffusion (as claimed in the reply to reviewer C5HF). Just as an example of a significant point of difference, conditioning is performed differently in these works - you use the decomposition of the posterior score into unconditional + conditioning term, whereas they just feed in the conditioning information into the architecture. Without a detailed side-by-side comparison, it is challenging to agree that Cohesion generalises these frameworks.
> >
> > If we choose to accept that Cohesion is a generalisation of the above-mentioned frameworks, I still believe that a comparison to them is needed.
> > - If Cohesion outperforms them, it only strengthens the paper, achieving better results with significantly reduced computational cost.
> > - If it doesn’t (which is my suspicion), one needs to ask why a more general framework is not able to converge to the same optimum as a less general framework. A plausible explanation could be that Cohesion incorporates weaker inductive biases, making it less efficient in the learning process.
> >
> > **Conclusion**
> > In conclusion, I greatly appreciate the extra empirical evidence and I find the new manner in which the authors propose to pose the contributions of the paper much more suitable. However, the message in the current manuscript is not exactly the same as the one outlined in this rebuttal, and it would require a significant amount of change to the current version to align them. Finally, I maintain my opinion that in order to better place the performance of the Cohesion framework in the context of diffusion-based PDE solvers, a comparison to other relevant (diffusion-based) baselines is needed. This includes both a metric comparison as well as a computation time comparison (where Cohesion would probably outperform).

---

### Official Review · Reviewer_L6gh · 2024-10-28

**Soundness:** 3
**Presentation:** 2
**Contribution:** 3
**Rating:** 3
**Confidence:** 3

**Summary:**

The paper introduces Cohesion, a framework for probabilistic dynamics forecasting in chaotic systems like fluid dynamics. By reframing forecasting as a trajectory-planning task, the framework makes use of reduced-order models (ROM) to make denoising processes more efficient. This approach is much faster because it applies a single denoising pass to the whole forecast sequence, rather than using multiple autoregressive steps. Cohesion also includes a way of guiding the process without using classifiers, which makes it suitable for zero-shot forecasting. Tests on the Kolmogorov Flow and Shallow Water Equation show that Cohesion works better than other methods for capturing multi-scale physical structures and reducing spectral divergence, which is important for modeling chaotic systems accurately.

**Strengths:**

The framework is innovative in its approach to combining turbulence theory, conditional generative model, and reinforcement learning principles for forecasting. By applying diffusion models with coherence-based conditioning, the paper contributes a novel perspective on long-range forecasting. The idea of Turbulence-diffusion framework and the induced Zero-shot conditional sampling is interesting and inspiring.  Cohesion presents a promising solution for long-range dynamics forecasting, especially in chaotic and partially observed environments. Experiments are exhaustive.

**Weaknesses:**

**Over-claims:**

**Reinforcement Learning:** After reading the methodologies, I didn't see RL play any role in this work. The only related part is borrowing the idea from Diffuser (Janneretal. 2022), to do the (sub) trajectory generation rather than autoregressive generation for efficiency. Such point and even claiming to achieve stable long rollouts with RL are invalid to me, given that Diffuser is a purely a generative method but to solve RL problems only. The critical components like **value function and reward function** never appear.


**Zero-shot:** While zero-shot forecasting is addressed conceptually, further clarification on how this aspect was validated experimentally is lacking.



**Writing:**

- Abstract: some sentences are naively extracted from the introduction but unclear after compression. "Nonetheless,Cohesionsupports...", unclear what's the advantage the authors refer to. Specifically, why iterations over subsequences are not allowed in the previous works. And direct abbreviation "NFEs".
- Section 2: The demonstration of $u_K$ and the relationship between $u_K$ and $u$ should be exposed clearer. E.g. change the title of "Coherent flow as conditioning prior" to "Conditional Diffusion Modeling", then put the demonstration of $u_K$ here first, and then introduce the coherent flow is the conditioning prior.

**Questions:**

1. **On Model Flexibility**: ROM is a type of method with limited expressiveness e.g. Koopman Operator, a linear approximation model. Would Cohesion be adaptable to domains where coherent flow cannot be efficiently approximated by ROM?
2. **Baselines:** SFNO seems the only baseline. How about others like Markov Neural Operator (MNO)?
3. **Long-Term Stability**: The results are over long rollouts, how long is it and how hard is it to predict that?
4. **Ablation Study** has been lacking, such as W window size.

---

> ### Author Response · Authors · 2024-11-23
>
> We would like to thank Reviewer L6gh for their constructive feedback! Please find our line-by-line response to the concerns raised.
>
> > The critical components like value function and reward function never appear.
>
> We agree that we draw inspiration from the Diffuser paper to solve long-range forecasting of nonlinear dynamical system. Although not explicitly demonstrated in the paper, we clarify in the future work section that additional reward function could easily be incorporated using Diffuser's framework, such as the use of physics-based constraints for stability guidance e.g., [1].
>
> > While zero-shot forecasting is addressed conceptually, further clarification on how this aspect was validated experimentally is lacking.
>
> We further clarified that all the experiments and ablations conducted here use only a _singly_ trained score network. This underscores the zero-shot forecasting concept formalized in the paper that remains independent of the conditioning prior.
>
> > Baselines: SFNO seems the only baseline. How about others like Markov Neural Operator (MNO)?
>
> We have added probabilistic formulation of UFNO and the classical (tensorized) FNO, all with identical parameter budget as our Cohesion framework. These models are implemented off-the-shelf from https://github.com/neuraloperator/neuraloperator. The table below summarizes the results after evaluating for the best ensembling strategies (IC-perturb / MC-dropout / Multi-checkpoints), across Kolmogorov/SWE experiments, at the final timestep T.
>
> |                 | **Cohesion (R=1)** | **Cohesion (R=T)** | **TFNO (ensemble)** | **UFNO (ensemble)** | **SFNO (ensemble)** |
> |-----------------|--------------------|--------------------|---------------------|---------------------|---------------------|
> | **RMSE (↓)**    | **0.31 / 0.25**    | **0.37 / 0.40**    | 0.40 / 1.52         | 0.42 / 0.92         | 0.49 / 0.93         |
> | **MAE (↓)**     | **0.25 / 0.15**    | **0.27 / 0.20**    | 0.30 / 1.32         | 0.35 / 0.51         | 0.40 / 0.52         |
> | **MS-SSIM (↑)** | **0.68 / 0.95**    | **0.45 / 0.90**    | 0.01 / 0.01         | 0.21 / 0.42         | 0.32 / 0.43         |
>
> __Conclusion 1: Cohesion outperforms all xNOs baselines__
>
> > On Model Flexibility: ROM is a type of method with limited expressiveness e.g. Koopman Operator, a linear approximation model. Would Cohesion be adaptable to domains where coherent flow cannot be efficiently approximated by ROM?
>
> Based on Koopman theory [2], any nonlinear PDEs can, in theory, be approximated by infinite-dimensional linear approximations, and so an ever-larger ROM can always be used to approximate chaotic system with high-dimensional invariances (e.g., attractors). Although we do agree that _infinite_ approximation here can be problematic in practice, we argue that the linearization conferred by Koopman operator allows us to perform stability analysis and identify situations where error growth becomes significant (high Lyapunov exponent) such that stability constraints can be imposed [3]. To underscore why stability is of paramount importance, we substitute deep Koopman operator, that both truncates and linearizes the PDE solution to mimic mean flow in Reynold's decomposition, with xNOs, and find substantial degradation at long-range rollouts.
>
> |                 | **Cohesion (TFNO)** | **Cohesion (UFNO)** | **Cohesion (SFNO)** | **Cohesion (ROM)** |
> |-----------------|---------------------|---------------------|---------------------|--------------------------|
> | **RMSE (↓)**    | 0.41 / >2.00        | 0.42 / >2.00        | 0.46 / 1.51         | **0.37 / 0.40**          |
> | **MAE (↓)**     | 0.30 / >2.00        | 0.31 / >2.00        | 0.36 / 0.65         | **0.27 / 0.20**          |
> | **MS-SSIM (↑)** | 0.01 / 0.30         | 0.27 / 0.33         | 0.39 / 0.50         | **0.45 / 0.90**          |
>
> __Conclusion 2: ROMs stabilize the reverse conditional denoising process, while remaining robust and skillful. Even in the case of nonlinear dynamics with high-dimensional invariant measures, an ever-larger linearized ROMs can be used at the benefits of tractability (linear stability analysis) and stability (coherence)__
>
>
> __References:__
>
> [1] Schiff, Yair, et al. "DySLIM: Dynamics Stable Learning by Invariant Measure for Chaotic Systems." arXiv preprint arXiv:2402.04467 (2024).
>
> [2] Koopman, Bernard O. "Hamiltonian systems and transformation in Hilbert space." Proceedings of the National Academy of Sciences 17.5 (1931): 315-318.
>
> [3] Chattopadhyay, Ashesh, and Pedram Hassanzadeh. "Long-term instabilities of deep learning-based digital twins of the climate system: The cause and a solution." arXiv preprint arXiv:2304.07029 (2023).

---

> > ### Comment · Reviewer_L6gh · 2024-11-26
> > **Response**
> >
> > I appreciate the effort the authors put to address my concerns. But I decide to keep my score, the reasons are as follows:
> > - Overclaims about RL. Getting inspiration from RL is fine, but then when the elements of RL is lacking, I personally don't think it is acceptable for the authors to bring RL up in the introduction. In fact, it is far from RL, and the inspiration should be more about sequence generation.
> > - Zero-shot. I'm not sure here it should be understood as a generalization ability or zero-shot. And lack of details about this in the experiments.
> > In conclusion, I don't think this work is ready to publish. The logic, inspiration, intuition, and some part of the experiments about demonstrating zero-shot ability, should be rewritten.

---

### Official Review · Reviewer_C5HF · 2024-11-03

**Soundness:** 3
**Presentation:** 2
**Contribution:** 2
**Rating:** 5
**Confidence:** 3

**Summary:**

The paper proposes Cohesion, a coherence-based diffusion model for long-range dynamics forecasting, aimed at addressing challenges in autoregressive probabilistic forecasting.

**Strengths:**

- The integration of turbulence and diffusion principles with ROM-based conditioning is novel and provides a new approach for multi-scale and chaotic systems.
- This paper uses quantitative (RMSE, MAE, MS-SSIM) and physics-based metrics (spectral divergence), provide strong empirical support.

**Weaknesses:**

- Lack of interpretability. How to prove the role of coherent flow after decomposition and how it promotes long-term stability prediction? Lack of theoretical explanation.

- Please explain what the zero-shot forecasts without classifier and multi-scale physical structure mentioned in the paper are. There exists confusion in the statements.

- Please explain why SFNO is used as the baseline. SFNO is mainly used to predict atmospheric dynamics. Intuitively, the datasets used in this paper are not suitable for spherical geometry operators. Please give an explanation.

- The baseline only uses SFNO. Why not compare other neural operator models, such as CNO[1], UNO[2], LSM[3], etc.

- The model is based on Diffusion. Why not compare diffusion-based models, such as PreDiff[4] and DYffusion[5].

- The time complexity comparison analysis of the model should be increased.


[1] Bogdan Raonic wt al. 'Convolutional Neural Operators for robust and accurate learning of PDEs.' NeurIPS2023.

[2] Md Ashiqur Rahman et al. 'U-NO: U-shaped Neural Operators.' TMLR2023.

[3] Haixu Wu et al. 'Solving High-Dimensional PDEs with Latent Spectral Models.' ICML2023.

[4] Zhihan Gao et al. 'PreDiff: Precipitation nowcasting with latent diffusion models.' NeurIPS2023.

[5] Salva Rühling Cachay et al. 'DYffusion: A Dynamics-informed Diffusion Model for Spatiotemporal Forecasting.' NeurIPS2023.

**Questions:**

Please address the questions in the Weaknesses.

---

> ### Author Response · Authors · 2024-11-23
>
> We would like to thank Reviewer C5HF for their constructive feedback. We address your concerns point-by-point by conducting additional experiments and/or providing further clarifications.
>
> > Please explain what the zero-shot forecasts without classifier and multi-scale physical structure mentioned in the paper are. There exists confusion in the statements.
>
> The term zero-shot forecast is primarily used to describe how, with just a singly-trained (unconditional) score network, we are able to estimate the posterior distribution without being tied to the conditioning factor during the reverse process. This then allows us to run all the experiments and ablations with just a single score network, without re-training new ones whenever the distribution of the conditioning factors shifts (i.e., the use of different ROM architecture e.g., SFNO, UFNO, FNO, Koopman).
>
> With regards to multi-scale physical structure, it primarily refers to the length scale of the dynamics e.g., from small-scale eddies to large-scale mean flow. An ideal PDE solver for fluid is able to capture all these across-scale dynamics.
>
> > The baseline only uses SFNO. Why not compare other neural operator models, such as CNO[1], UNO[2], LSM[3], etc.
>
> We agree that the use of SFNO with no spherical geometry is less ideal, which is the case for Kolmogorov flow where the dynamics is discretized along non-spherical grid. As such, we have added probabilistic formulation of UFNO and the classical (tensorized) FNO, all with identical parameter budget as our Cohesion framework. The table below summarizes the results after evaluating for the best ensembling strategies (IC-perturb / MC-dropout / Multi-checkpoints), across Kolmogorov/SWE experiments, at the final timestep T.
>
> |                 | **Cohesion (R=1)** | **Cohesion (R=T)** | **TFNO (ensemble)** | **UFNO (ensemble)** | **SFNO (ensemble)** |
> |-----------------|--------------------|--------------------|---------------------|---------------------|---------------------|
> | **RMSE (↓)**    | **0.31 / 0.25**    | **0.37 / 0.40**    | 0.40 / 1.52         | 0.42 / 0.92         | 0.49 / 0.93         |
> | **MAE (↓)**     | **0.25 / 0.15**    | **0.27 / 0.20**    | 0.30 / 1.32         | 0.35 / 0.51         | 0.40 / 0.52         |
> | **MS-SSIM (↑)** | **0.68 / 0.95**    | **0.45 / 0.90**    | 0.01 / 0.01         | 0.21 / 0.42         | 0.32 / 0.43         |
>
> __Conclusion 1: Cohesion outperforms all xNOs baselines__
>
> > Lack of interpretability. How to prove the role of coherent flow after decomposition and how it promotes long-term stability prediction? Lack of theoretical explanation.
>
> In order to illustrate this point and draw the empirical connection with Reynold's decomposition, we perform two additional ablations: (1) substituting deep Koopman operator with TFNO/UFNO/SFNO, and (2) analyzing the scaling property of the ROM. Both analysis are performed on the trajectory mode (R=T) for Kolmogorov/SWE problems, at the final timestep T.
>
> __(1) Choice of mean flow estimator__: We substitute deep Koopman operator, that both truncates and linearizes the PDE solution to mimic mean flow in Reynold's decomposition, with xNOs, and find substantial degradation at long-range rollouts.
> |                 | **Cohesion (TFNO)** | **Cohesion (UFNO)** | **Cohesion (SFNO)** | **Cohesion (ROM)** |
> |-----------------|---------------------|---------------------|---------------------|--------------------------|
> | **RMSE (↓)**    | 0.41 / >2.00        | 0.42 / >2.00        | 0.46 / 1.51         | **0.37 / 0.40**          |
> | **MAE (↓)**     | 0.30 / >2.00        | 0.31 / >2.00        | 0.36 / 0.65         | **0.27 / 0.20**          |
> | **MS-SSIM (↑)** | 0.01 / 0.30         | 0.27 / 0.33         | 0.39 / 0.50         | **0.45 / 0.90**          |
>
> __Conclusion 2: ROMs stabilize the reverse conditional denoising process, while remaining robust and skillful__
>
> __(2) Scaling property of ROM__: We ablate the ROM with orders-of-magnitude reduction in parameter size. Even with 2 orders magnitude smaller ROM, Cohesion is competitive even with the best probabilistic version of xNOs.
>
> |                 | **Cohesion (x$10^{-2}$)** | **Cohesion (x$10^{-1}$)** | **Cohesion (x$10^0$)** | **Reference (best xNO)** |
> |-----------------|---------------------------|---------------------------|------------------------|--------------------------|
> | **RMSE (↓)**    | 0.39 / 0.95               | 0.38 / 0.90               | 0.37 / 0.40            | 0.40 / 0.92              |
> | **MAE (↓)**     | 0.30 / 0.50               | 0.28 / 0.49               | 0.27 / 0.20            | 0.30 / 0.51              |
> | **MS-SSIM (↑)** | 0.27 / 0.59               | 0.40 / 0.60               | 0.45 / 0.90            | 0.32 / 0.43              |
>
> __Conclusion 3: Diffusion scales with coarse flow approximation__

---

> ### Author Response · Authors · 2024-11-23
>
> > Diffusion baseline
>
> With regards to comparison with other diffusion methods, like Dyffusion we argue that this is similar to our Cohesion framework in autoregressive mode (R=1) at which correction is performed, conditional on previous forecast. In this case, we find substantial increase in inference speed, if we perform inference as trajectory planning (R=T), with minimal deterioration in skill. In a sense, our framework provides a __generalization__ to how forecasting is performed with e.g., Dyffusion, as a special case of R=1, video generation task as R=T, and anywhere in between as R=[1..T], for a more flexible design and provides strategies (model-free temporal convolution) to ensure consistency.

---

### Meta-Review · Area_Chair_AwLM · 2024-12-22

**Metareview:**

The paper focuses on long-range forecasting of dynamics with a “trajectory planning” perspective. It recognizes a parallel between diffusion-based models and turbulence. Specifically, it conditions a diffusion process on a prior of coherent structures obtained from reduced order modeling of all snapshots of a sequence and generates the whole dynamics as a single denoising process with a reconstruction guidance. It tests the method on the two chaotic systems of Kolmogorov flow and shallow water and compares favorably with some included baselines.

The reviewers appreciated the novelty of the ROM based prior distribution, the efficient non-autoregressive generation, the significant empirical support with several physical and image-based metrics,

On the other hand, they raised important concerns regarding similarity of contributions to prior work without proper description of the differences (particularly Rozet et al.), lack of details on different components of the method design and experiments, lack of several relevant baselines including neural operators and diffusion based models among others, and potential overclaiming regarding parallels to reinforcement learning despite of no empirical evidence of a full RL setup.

The authors provided a thorough rebuttal where they included some more baselines and clarified some of the uncertainties about the details of the methods, the experimental setup, and the rationale for the selection of the current baselines.

Most reviewers attended to the rebuttal but did not find it convincing. The main outstanding concerns are proper reference of prior work and what is the precise contribution of this work and the inclusion of several relevant baselines that are currently absent from this work.

All reviewers eventually recommend rejection with which the AC agrees. The paper should undergo a major revision not only to address the above two main concerns but also several relevant and detailed feedback from the five expert reviewers.

**Additional Comments On Reviewer Discussion:**

Five expert reviewers evaluated the paper with expertise in physics, generative modeling, and reinforcement learning. They provided detailed review and mostly considered the rebuttal. The reviewers unanimously lean towards rejection.

---

### Decision · Program_Chairs · 2025-01-22

Reject